# Evaluation of Four Atmospheric Correction Algorithms for GOCI Images over the Yellow Sea

**Xiaocan Huang [1],*, Jianhua Zhu [1] , Bing Han [1] , Cédric Jamet [2] , Zhen Tian [1] , Yili Zhao [1], Jun Li [1] and Tongji Li [1]**

1 National Ocean Technology Center, State Oceanic Administration, Tianjin 300112, China
2 Univ. Littoral Cote d'Opale, CNRS, Univ. Lille, UMR 8187, F 62930 Wimereux, France
* Correspondence: huangxiaocan16@mails.ucas.ac.cn; Tel.: +86-22-2753-6515

**Abstract:** Atmospheric correction (AC) for coastal waters is an important issue in ocean color remote sensing. AC performance is fundamental in retrieving reliable water-leaving radiances and then bio-optical parameters. Unlike polar-orbiting satellites, geostationary ocean color sensors allow high-frequency (15–60 min) monitoring of ocean color over the same area. The first geostationary ocean color sensor, i.e., the Geostationary Ocean Color Imager (GOCI), was launched in 2010. Using GOCI data acquired over the Yellow Sea in summer 2017 at three principal overpass times (02:16, 03:16, 04:16 UTC) with ±1 and ±3 h match-up times, this study compared four GOCI AC algorithms: (1) the standard near infrared (NIR) algorithm of NASA (NASA-STD), (2) the Korea Ocean Satellite Center (KOSC) standard algorithm for GOCI (KOSC-STD), (3) the diffuse attenuation coefficient at 490 nm Kd (490)-based NIR correction algorithm (Kd-based), and (4) the Management Unit of the North Sea Mathematical Models (MUMM). The GOCI-estimated remote sensing reflectance (Rrs), aerosol parameters [aerosol optical thickness (AOT), Angström Exponent (AE)], and chlorophyll-a (Chla) were validated using in situ data. For Rrs, AOT, AE, and Chla, GOCI-retrieved results performed well within the ±1 h temporal window, but the number of match-ups was extended within the ±3 h match-up window. For ±3 h GOCI-derived Rrs, all algorithms had an absolute percentage difference (APD) at 490 and 555 nm of <40%, while other bands showed larger differences (APD > 60%). Compared with in situ values, the APD of the Rrs(490)/Rrs(555) band ratio was <20% for all ACs. For AOT and AE, the APD was >40% and >200%, respectively. Of the four algorithms, the KOSC-STD algorithm demonstrated satisfactory performance in deriving Rrs for the region of interest (Rrs APD: 22.23%–73.95%) in the visible bands. The Kd-based algorithm worked well obtaining Ocean Color 3 GOCI Chla because Rrs(443) is more accurate than the KOSC-STD. The poorest Rrs retrievals were achieved using the NASA-STD and the MUMM algorithms. Statistical analysis indicated that all methods had optimal performance at 04:16 UTC.

**Keywords:** ocean color; atmospheric correction; GOCI; Yellow Sea; validation; aerosols; remote sensing reflectance; chlorophyll-a

---

## 1. Introduction

Ocean color remote sensing with daily–hourly sampling frequency and broad spatial coverage plays a critical role in the investigation of the bio-optical properties and the biogeochemical parameters of nearshore coastal waters. Although they represent only 7% of the total ocean surface, coastal and inland waters produce up to 40% of marine and freshwater biomasses inventoried today and 85% of marine and freshwater resources exploited by humans. Moreover, 60% of the world's population lives within 100 km of a coast, whilst inland waters provide key ecosystem services with direct linkages to human health [1]. These waters, which are very often optically complex, are generally identified as

Case-2 waters [2]. The optical complexity of these waters is due to the presence of particles other than phytoplankton, i.e., suspended sediments and colored dissolved organic matter (CDOM), and it can exhibit high diurnal dynamics [1].

In the past two or more decades, various spaceborne ocean color sensors have been launched to perform global observations at daily–yearly timescales. While the past and the present primary observing platforms comprise polar orbiting satellites [i.e., the Sea-Viewing Wide Field-of-View Sensor (SeaWiFS), the Moderate Resolution Imaging Spectroradiometer (MODIS), the Medium Resolution Imaging Spectroradiometer (MERIS), the Visible Infrared Imaging Radiometer Suite (VIIRS), and the Ocean Land Colour Instrument (OLCI)], the world's first Geostationary Ocean Color Imager (GOCI), launched by South Korea in 2010, represented a major breakthrough. It was designed for oceanic applications over open and coastal waters, and it observes the same area with high-temporal frequency (an image every hour and up to eight per day) and spatial resolution of 500 m [3].

To estimate the optical and the biogeochemical properties of coastal waters, the atmospheric contribution should be removed first from the total signal measured by the spaceborne sensor. Generally, about 90% of a signal originates from atmospheric effects, while the upwelling radiance emerging from the water surface contributes <10% [4,5]. The process adopted to remove the major atmospheric signals and to extract the minor water signals, i.e., the water-leaving radiance (Lw) or the remote sensing reflectance (Rrs), is called atmospheric correction (AC). Over open ocean waters, the "black-pixel" assumption is applied in this process [6] (hereafter GW-AC). This hypothesis considers that Lw is zero in the near infrared (NIR) bands because of the strong absorption of the water itself. However, the black-pixel assumption is invalid for turbid waters [7,8], leading to significant errors in retrieving ocean color products [4,9,10]. To overcome this problem, several specific AC algorithms have been developed that account for the non-negligible contribution of Lw in the NIR bands for Case-2 waters. These algorithms can be grouped into five categories: (1) assignment of hypothesis on the NIR aerosols or water contributions [11–13], (2) use of shortwave IR bands [14–16], (3) use of blue or UV bands [17,18], (4) correction/modeling of the non-negligible ocean in the NIR [7,19,20], and (5) coupled ocean/atmosphere inversion based on artificial neural networks [21,22] or optimization techniques [23–25]. Extensive validation of these methods is required to assess their accuracy and applicability over coastal waters with different optical properties. Previously, considerable research was undertaken on regional evaluation of AC for polar-orbiting ocean color sensors (e.g., [7,11,26]); however, few studies [27] have compared the AC algorithms for GOCI using in situ data specifically over Chinese coastal waters. Therefore, this was the focus of our research.

In this study, GOCI images were processed using four AC algorithms: (1) the NASA standard AC algorithm using an iterative procedure and a bio-optical model [28] (hereafter, NASA-STD); (2) the Korea Ocean Satellite Center (KOSC) standard algorithm based on iterative models using relationships between the red and the NIR bands [29] (hereafter, KOSC-STD); (3) the iterative NIR correction algorithm based on a regional empirical relationship between the NIR nLw(λ) and diffuse attenuation coefficient at 490 nm [Kd(490)] (hereafter, Kd-based) [30]; and (4) the modification of NIR correction model assuming spatial homogeneity of the aerosol reflectance and Lw in the NIR bands over the region of interest [12] (hereafter, MUMM). The first three algorithms belong to the fourth category AC algorithm mentioned above, while the fourth algorithm belongs to the first category.

This paper is organized as follows. Section 2 describes the four different algorithms applied to the GOCI images used in this study. The data and the methods used in the research are introduced in Section 3. In this section, the characteristics of in situ and satellite data are outlined, and the match-up criteria as well as the evaluation indicators are described in brief. In Section 4, the comparison results for Lw and aerosol optical parameters retrieved by the different algorithms at three GOCI overpass times (02:16, 03:16, 04:16 UTC) for ±3 h and ±1 h are presented. The final two sections not only discuss the performance and the limitations of the AC algorithms, but they also offer suggestions and perspectives regarding further AC improvements.

## 2. Algorithms

### 2.1. AC Algorithms

The total reflectance ($\rho_t$) measured by the GOCI sensor at the top of the atmosphere (TOA) can be expressed as [4]:

$$\rho_t(\lambda) = \rho_r(\lambda) + \rho_A(\lambda) + T(\lambda)\rho_g(\lambda) + t_v(\lambda)\rho_{wc}(\lambda) + t_v(\lambda)\rho_w(\lambda), \tag{1}$$

where $\rho_r(\lambda)$ represents the Rayleigh scattering radiance by air molecules; $\rho_A(\lambda)$ represents aerosol scattering, including the interaction between molecules and aerosols; $T(\lambda)$ is the direct transmittance between the sea surface and the TOA along the viewing direction; $\rho_g(\lambda)$ is the sunglint radiance; $t_v(\lambda)$ is the diffuse transmittance through the atmosphere; $\rho_{wc}(\lambda)$ represents the contribution from the whitecaps; and $\rho_w(\lambda)$ is the desired Lw. The purpose of AC is to remove the atmospheric and the surface contributions to determine $\rho_w(\lambda)$. When $\rho_t(\lambda)$ is corrected for gas absorption, Rayleigh scattering, sunglint, and whitecaps [31–33], the remaining terms can be denoted as the Rayleigh-corrected reflectance, $\rho_{rc}(\lambda)$:

$$\rho_{rc}(\lambda) = \rho_t(\lambda) - \rho_r(\lambda) - T(\lambda)\rho_g(\lambda) - t_v(\lambda)\rho_{wc}(\lambda) = \rho_A(\lambda) + t_v(\lambda)\rho_w(\lambda). \tag{2}$$

The four AC algorithms used in this study are briefly described in the following subsections. They each extend the GW-AC approach to derive Lw from TOA reflectance for turbid waters.

### 2.1.1. NASA Standard Algorithm (NASA-STD)

The NASA-STD AC algorithm was initially developed by Gordon and Wang [6], extended for application to turbid waters by Stumpf et al. [19], and revised by Bailey et al. and Ahmad et al. [28,34]. The latter revision is used by default in the present SeaDAS package version 7.X. The algorithm is based on an iterative process that accounts for the non-zero $\rho_w$ in the NIR bands. This method uses 80 aerosol models built from AERONET observations and vector radiative transfer code for the ocean-atmosphere system [34]. First, the black-pixel assumption is adopted using GW-AC to retrieve $\rho_w$ at 443 and 555 nm. Next, these two $\rho_w$ values are used as input for a bio-optical model (standard OBPG Chla OC3 algorithm [35], formula (1) in Figure 1) to obtain initial estimates of the chlorophyll-a (Chla) concentration, which then makes further efforts to determine particulate and CDOM absorption in the red band (formula (2) in Figure 1), a(660). Then, a(660) and $\rho_w$(660) are used to compute the particulate backscattering in the red band, b$_{bp}$(660), following which, b$_{bp}$(NIR) can be estimated in accordance with a power exponent function [36] (formula (3) in Figure 1). Therefore, values of $\rho_w$ in the NIR bands (or equivalently Rrs) are generated on the basis of these three relationships, and they are removed from $\rho_{rc}$(NIR). Ultimately, the procedure is repeated until convergence is reached. NASA uses this algorithm to generate its official GOCI L2 products.

### 2.1.2. KOSC Standard Algorithm (KOSC-STD)

The KOSC-STD algorithm is a modification of the GW-AC method, which additionally includes an iterative procedure to correct NIR ocean reflectance. For the NIR correction model, the same equations are used in GDPSv2.0 (GOCI Data Processing System, version2.0) and in GDPSv1.3 [29]. An empirical relationship is used to express the normalized water-leaving reflectance ($\rho_{wn}$) in the NIR band (745 nm) from $\rho_{wn}$(660) (formula (1) in Figure 1):

$$\rho_{wn}(745) = \sum_{n=1}^{6} j_n \rho_{wn}^n(660). \tag{3}$$

The nonlinear polynomial model between the two NIR bands (745 and 865 nm) is calculated as follows (formula (2) in Figure 1):

$$\rho_{wn}(865) = \sum_{n=1}^{2} k_n \rho_{wn}^n(745),$$ (4)

where $j_n$ ($n = 1, 2, \dots, 6$), $k_n$ ($n = 1, 2$) are known fitting coefficients [37].

### 2.1.3. Kd-Based NIR Correction Algorithm (Kd-Based)

Wang et al. [13] developed an algorithm specifically for processing GOCI ocean color data acquired in the highly turbid western Pacific region, including the Bohai Sea, the Yellow Sea, and the East China Sea. It is based on the regional relationship between nLw(745), nLw(865), and the diffuse attenuation coefficient at 490 nm [Kd(490)], which is derived from long-term MODIS-Aqua measurements (2002–2009) using NIR-based ocean color data processing (formula (1) in Figure 1) [38]:

$$nLw(745) = c_1 K_d(490) + c_2 K_d(490)^2 + c_3 K_d(490)^3 + c_4 K_d(490)^4,$$ (5)

where $c_1 = 0.465$, $c_2 = -0.385$, $c_3 = 0.152$, and $c_4 = -0.0121$. Similarly, nLw(865) can be formulated as (formula (2) in Figure 1):

$$nLw(865) = b_1 nLw(745) + b_2 nLw(745)^2,$$ (6)

where $b_1 = 0.368$ and $b_2 = 0.040$. Then, an iterative process is conducted to calculate nLw(745) and nLw(865) with the inputs of Kd(490) (Figure 1).

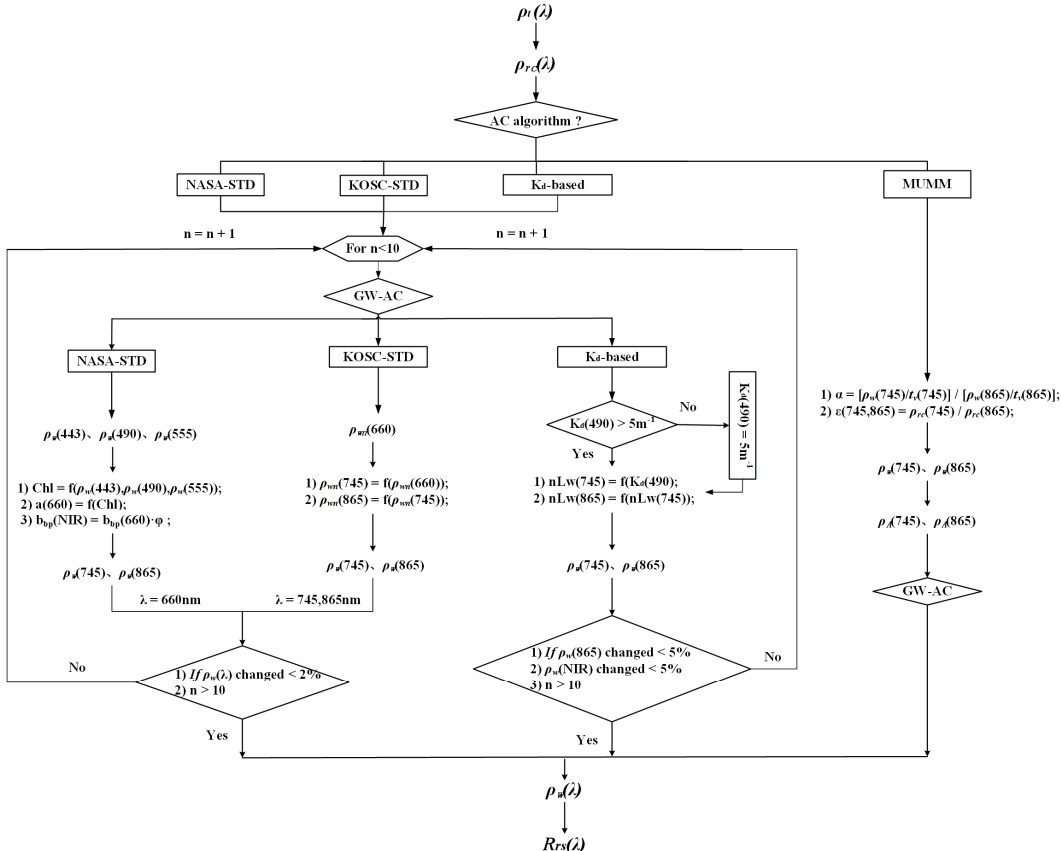

**Figure 1.** Schematic flowchart of the standard near infrared (NIR) algorithm of NASA (NASA-STD), the Korea Ocean Satellite Center (KOSC-STD), the diffuse attenuation coefficient at 490 nm Kd (490)-based NIR correction algorithm (Kd-based), and the Management Unit of the North Sea Mathematical Models (MUMM) algorithms. NIR denotes both Geostationary Ocean Color Imager (GOCI) 745 nm and 865 nm.

### 2.1.4. MUMM Algorithm

The fourth algorithm uses an analytical method that is referred to as the Management Unit of the North Sea Mathematical Models (MUMM) [30]. This algorithm replaces the invalid black-pixel assumption in the NIR bands over coastal waters with two alternative assumptions—one concerns the water optical properties, and the other concerns the atmosphere.

The first assumption originates from the fact that the shape of the NIR water spectrum is dominated primarily by pure water absorption and hence is invariant, while the magnitude of the signal is approximately proportional to the backscatter coefficient [30,39], which can be regarded as constant in the region of interest. The ratio of any two values of NIR water-leaving reflectance, named $\alpha$, is defined as (formula (1) in Figure 1):

$$\alpha = \frac{\rho_w(745)/t_v(745)}{\rho_w(865)/t_v(865)} \tag{7}$$

The second assumption arises from the fact that the aerosol concentration does not usually vary over spatial scales of about 100 km. Therefore, the ratio of multiple-scattering aerosol reflectance $\rho_a(\lambda) + \rho_{ra}(\lambda)$, named $\varepsilon$, can be considered constant over the region of interest. For turbid waters with clear waters nearby, $\varepsilon$ can also be calculated using the values of $\rho_{rc}(\lambda)$ in the clear waters as follows (formula (2) in Figure 1):

$$\varepsilon(745, 865) = \frac{\rho_{rc}(745)}{\rho_{rc}(865)} \tag{8}$$

One of the key points for the MUMM algorithm is to determine the values of $\alpha$ and $\varepsilon$. Here, $\alpha$ is set to 1.932 in accordance with Ruddick et al. [30]. To estimate $\varepsilon$(745,865) for each image, we produced scatterplots of $\rho_{rc}(745)$ versus $\rho_{rc}(865)$ for the region of interest, and we calculated the slope of the line for values of $\rho_{rc}(865) < 0.015$. Using the set value of $\alpha$ and the estimated values of $\varepsilon$, the following equations are defined:

$$\rho_A(865) = \frac{\alpha\rho_{rc}(865) - \rho_{rc}(745)}{\alpha - \varepsilon(745, 865)}, \tag{9}$$

$$\rho_A(745) = \varepsilon(745, 865)\left(\frac{\alpha\rho_{rc}(865) - \rho_{rc}(745)}{\alpha - \varepsilon(745, 865)}\right) \tag{10}$$

The values of $\rho_A(745)$ and $\rho_A(865)$ are estimated using Equations (9) and (10) to select appropriate aerosol models. Finally, the determined aerosol models are reentered into the GW-AC scheme.

### 2.2. Chla Retrievals

The accuracy of AC algorithms determines the precision of the extraction of ocean color parameters (e.g., Chla). To retrieve Chla, different types of algorithm (e.g., OC2, OC3, OC4, and GSM01) were adopted in previous studies [40–43]. In this study, considering the specific spectral bands of the GOCI sensor, Chla concentrations were derived from all atmospherically corrected images using the OC3G algorithm (Ocean Color 3 GOCI) [37]. OC3G is an empirically derived algorithm developed as an extension of OC4 and OC2 [43]. The general form of the OC3G algorithm can be expressed as:

$$Chla = 10^{f_0 + f_1 \cdot R + f_2 \cdot R^2 + f_3 \cdot R^3 + f_4 \cdot R^4} \tag{11}$$

$$R = \log_{10}\left(\frac{\max(Rrs(443), Rrs(490))}{Rrs(555)}\right) \tag{12}$$

with constant coefficients $f_0 = 0.0831$, $f_1 = -1.9941$, $f_2 = 0.5629$, $f_3 = 0.2944$, and $f_4 = -0.5458$. These coefficients are used in GPDSv2.0.

## 3. Data and Methods

### 3.1. Study Area

The Yellow Sea (YS) of China is the largest marginal sea of the northwestern Pacific Ocean (Figure 2). The YS lies within a shallow basin (average water depth: ~44 m) that is bounded by the Chinese mainland to the west and the Korean Peninsula to the east [44]. It covers an area of 417,000 km$^2$ [45]. One of the main characteristics of the YS is that the sea temperature and the salinity have prominent spatiotemporal diurnal variations [46,47]. The hydrologic and the circulation processes are governed by monsoon wind systems and the Kuroshio Current [30]. Because of the effects of these dominant processes, the Cold Water Mass, which is an important phenomenon within the YS, is formed with characteristics of low temperature (5–12 °C) and high salinity (31.5–32.5 psu) beneath a strong pycnocline [48], which is more prevalent during summer. Anthropogenic inputs such as pollution, eutrophic materials, and substantial sediment transport from the Yangtze River mean the YS region is well known for its high turbidity [49].

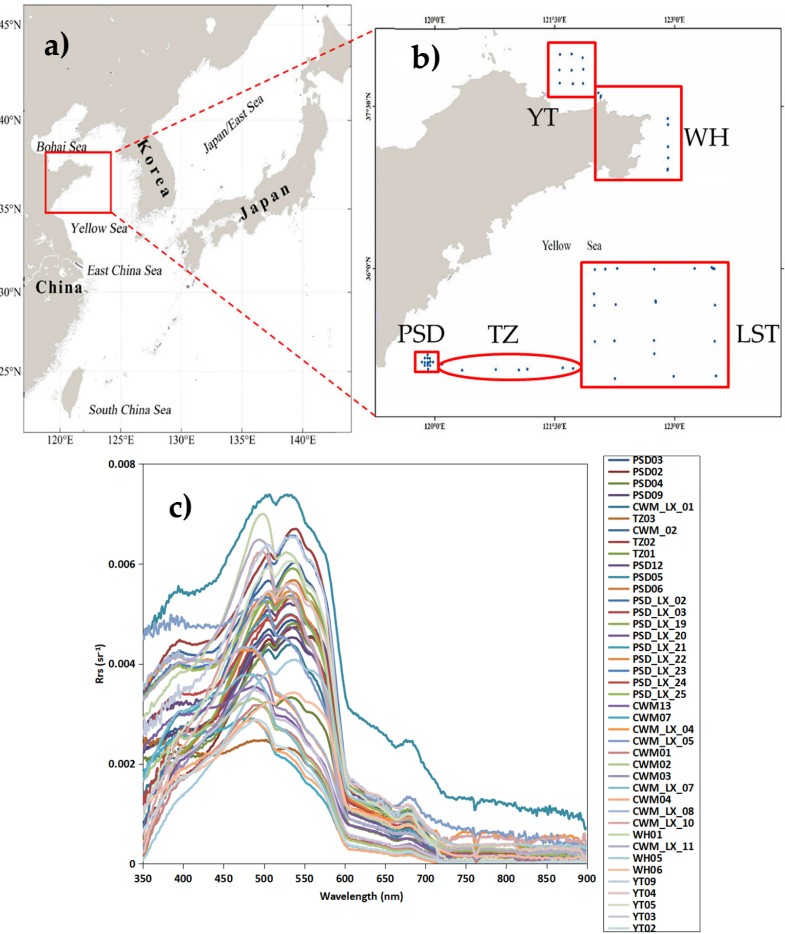

**Figure 2.** (**a**) Map of the region of interest, including the Bohai Sea, the Yellow Sea (YS), the East China Sea, the Japan/East Sea, and part of the South China Sea. The red box identifies the region over the YS used to analyze the temporal variability of atmospheric and water optical parameters measured by GOCI. (**b**) Locations of in situ data acquisition in the Yellow Sea in August 2017 are marked in solid blue circles. Five subregions that represent the different field data are marked where PSD represents PingShan Island, CWM denotes the Cold Water Mass, and TZ stands for the transitional zone between them. WH refers to the region over Weihai, and YT represents Yantai. For the spectra of remote sensing reflectance (Rrs) for each subregion, please refer to the Appendix A. (**c**) Spectra of Rrs at all 18 daytime stations.

*3.2. In Situ Measurements*

In situ bio-optical and atmospheric optical parameters were acquired during a sea cruise in the YS during 8–26 August 2017, which was conducted by the National Ocean Technology Center and the National Satellite Ocean Application Service. The data fields are exhibited in Figure 2. Overall, measurements were taken at 92 stations. Observations of remote sensing reflectance, atmospheric optical properties, and significant water constituents were acquired during the field experiment.

### 3.2.1. Measurement of Remote Sensing Reflectance

At stations with suitable conditions of solar illumination (generally between 09:00 and 15:00 local time), above-water optical measurements were conducted using a field spectrophotometer (ASD FieldSpec®3, full range: 350–2500 nm). The total upwelling radiance from the surface water ($L_t$), downwelling irradiance above the water surface ($E_s$), and downward sky radiance ($L_{sky}$) were measured. To avoid sunglint contamination, the zenith and the azimuth angles used to observe Lt were about 40° and 135° (referring to the solar plane), respectively. Furthermore, we selected the optimal orientation to minimize the influence of ship shading and whitecaps. Then, the remote sensing reflectance (Rrs, sr$^{-1}$) was calculated as follows [50]:

$$\begin{cases} L_{w+}(\lambda) = L_t(\lambda) - \rho_s(\lambda)L_{sky}(\lambda) \\ R_{rs}(\lambda) = L_{w+}(\lambda)/E_s(\lambda) \end{cases} \tag{13}$$

where $L_{w+}$ is the Lw just above the sea surface, and $\rho_s$ is the Fresnel reflectance at the air–water interface, which depends on viewing and illumination geometry, wind speed, cloud, and wavelength [51,52]. For this study, $\rho_s(\lambda)$ was retrieved using the nonlinear spectral optimization method and a bio-optical model for each station [53,54].

### 3.2.2. Measurement of Aerosol Optical Properties

Aerosol optical thickness (AOT) is defined as the integrated extinction coefficient over a vertical column of unit cross-section. It is a proxy for the concentration of aerosols within the air column. To validate the results of aerosol optical properties derived from the four AC algorithms, shipborne AOT was measured using a hand-held Microtops II sun photometer (SolarLight, USA) at five central wavelengths (i.e., 440, 500, 675, 870, and 1020 nm), denoted as AOT(440), AOT(500), AOT(675), AOT(870), and AOT(1020), respectively (Figure 3). For convenience of evaluation between satellite and field data, spline interpolation of AOT(500) and AOT(675) was applied to obtain AOT(555). The method for the retrieval of AOT from direct solar irradiance measurements is described by [50,55]. For details regarding cloud-screening and quality control procedures, the reader is referred to [56]. Additionally, the Angström exponent, AE, is commonly used to provide basic information on the particle size distribution and the type of aerosols. In this study, in situ AE between 440 and 870 nm, i.e., AE(440,870), was determined using linear regression with log-transformed in situ AOT(440) and AOT(870) as follows:

$$\text{AE}(440,870) = -\frac{In[AOT(440)/AOT(870)]}{In(440/870)} \tag{14}$$

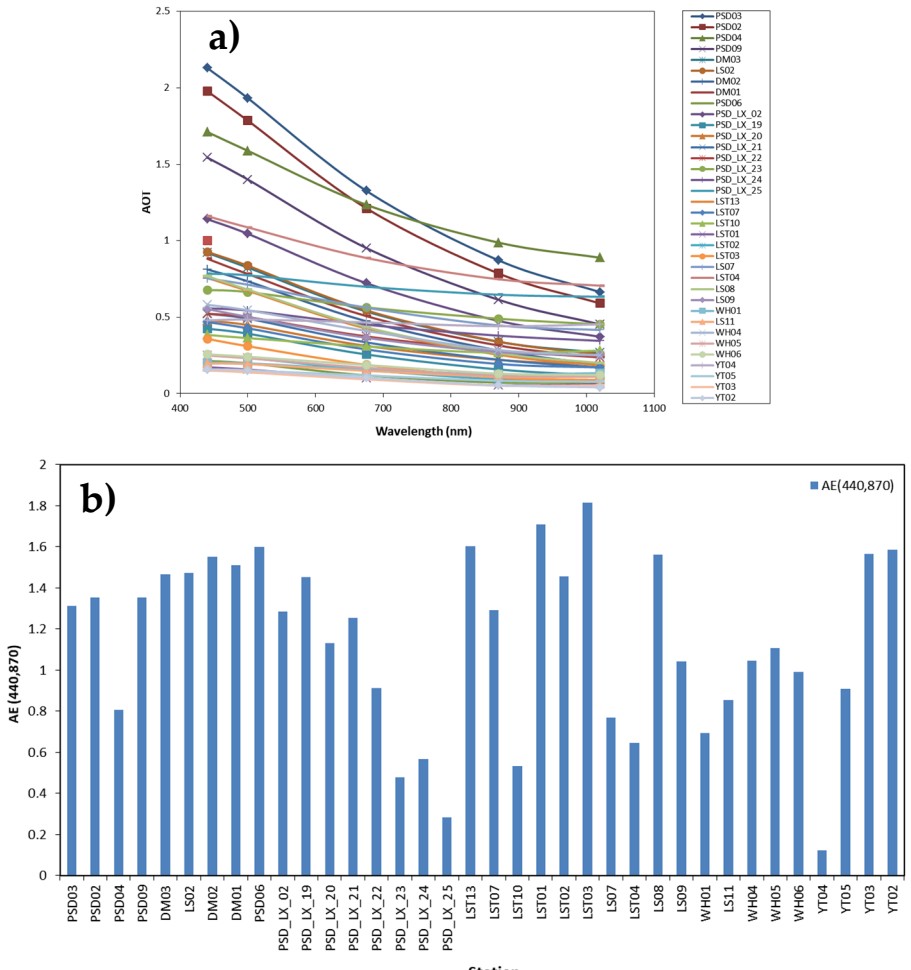

**Figure 3.** (**a**) Spectra of in situ aerosol optical thickness (AOT). (**b**) Histogram of in situ Angström Exponent (AE)(440,870) at 18 daytime stations.

### 3.2.3. Measurement of Chla Data

In situ surface Chla concentrations were used for validation of the imagery-derived Chla products of the four AC algorithms. Chla samples (n = 92) were collected at the surface using a continuous shipboard laboratory pump. The surface water samples were filtered using 0.7 μm Whatman GF/F glass fiber filters following Ocean Optics protocols [57]. Chla samples were stored at −20 °C until the samples were processed. Chla pigment concentrations were extracted using a high-performance liquid chromatography (HPLC) system. The derived mean values and the standard deviations of in situ Chla were 0.78 and 0.51, respectively.

### 3.3. GOCI Data

The Geostationary Ocean Color Imager (GOCI), the world's first geostationary ocean color spaceborne instrument, was launched in June 2010 and provides eight images per day. It is one of the three payloads onboard the Communication, Ocean and Meteorological Satellite (COMS).

The GOCI instrument was designed to provide hourly data in eight bands in the visible and the NIR parts of the spectrum (412–865 nm) with 500 m spatial resolution. The region of GOCI observations, which covers an area of 2500 km × 2500 km (21.54°–46.99°N, 116.41°–148.67°E, centered at 36°N, 130°E), includes the coast of East China, the Korean Peninsula, and Japan [Figure 2a] [58].

The solar and the satellite viewing zenith angles of GOCI are reasonably small (<30°) between 02:16 and 04:16 UTC; therefore, available daily GOCI L1B images during this period and corresponding to the days of the sea cruise were downloaded from KOSC (http://kosc.kiost.ac.kr/eng/p30/kosc_p34.html).

The GOCI images were processed from L1B to L2 for the KOSC algorithms using GDPSv2.0 and using the SeaDAS software package version 7.5 (SeaDASv7.5, OBPG http://oceancolor.gsfc.nasa.gov/) for NASA and MUMM algorithms.

Here, the data analysis focuses on Rrs at 412, 443, 490, 555, 660, 680, 745, and 865 nm, AOT at 443, 555, 680, and 865 nm, and AE(443,865). Thus, the slight differences in the spectral channels between the GOCI images and the in situ data are ignored.

### 3.4. Match-Ups Procedures

Match-ups between the in situ and the GOCI-retrieved AOT and Rrs were selected based on locations and overpass times. The slight difference in the wavelengths between the in situ and the satellite-retrieved values was ignored. The match-up criteria adopted were similar to the approach described in Bailey and Werdell [59]. Match-up time-windows of ±1 h and ±3 h were compared for AOT, AE, Rrs, and Chla. First, 3 × 3 pixel boxes were extracted from the GOCI images centered on the measurement sites. Second, a coefficient of variation (CV; standard deviation divided by mean values) was calculated for each band to account for the spatial homogeneity of the pixels within each 3 × 3 box. Match-ups with CV values >0.2 in the 3 × 3 pixel boxes for Rrs(555) were excluded. We required that at least five valid pixels within the 3 × 3 pixel box be valid. Finally, the mean value of the remaining pixels was calculated.

### 3.5. Statistical Metrics

Statistical metrics were used to evaluate the four algorithms performance between in situ Microtops II aerosol products versus GOCI aerosol products, in situ Rrs versus GOCI Rrs products, and in situ Chla versus GOCI Chla products. Statistical parameters included the absolute percentage difference (APD, Equation (14)), the root mean square error (RMSE, Equation (15)), the Bias (Equation (16)), the correlation coefficient ($R^2$), and the slope and the intercept of the linear regression:

$$\text{APD} = \frac{1}{N} \sum_{i=1}^{N} \left| \frac{Y_i - X_i}{X_i} \right| \times 100\%, \tag{15}$$

$$\text{RMSE} = \sqrt{\frac{1}{N} \sum_{i=1}^{N} (Y_i - X_i)^2}, \tag{16}$$

$$\text{Bias} = \frac{1}{N} \sum_{i=1}^{N} (Y_i - X_i), \tag{17}$$

where $X_i$ is the i-th in situ observation, $Y_i$ is the i-th GOCI observation, and N is the number of match-ups between the in situ measurements and the GOCI-retrieved values.

These metrics represent unbiased statistics that reflect how accurately the GOCI-derived values agree with the field data. Both APD and RMSE are sensitive to outliers, Bias indicates the deviation level of the GOCI-retrieved values from the in situ data, and the $R^2$ value reflects the linear consistency between the in situ data and the GOCI measurements, which is related to data distributions. For this study, statistical significance was defined at the 95% confidence level.

## 4. Results

The criterion for GOCI cloud masking is different in GDPS than in SeaDAS [60]. Minor differences in flags between GDPS and SeaDAS mean the number of match-up pairs varies among the AC algorithms. We analyzed time-windows of ±1 and ±3 h for Rrs, AOT, AE, and Chla between the shipborne data and the GOCI overpass to choose the best time-window. Within a ±3 h match-up window, of the 78 available in situ Rrs measurements, 30, 29, 31, and 32 match-ups for Rrs(555) were available for the NASA-STD, the KOSC-STD, the Kd-based, and the MUMM algorithms, respectively.

The number of match-ups for each algorithm was greater with a time-window of ±3 h than with a time-window of ±1 h (i.e., 10, 10, 10, and 11 match-ups, respectively). The number of match-ups for AOT with ±3 and ±1 h time-windows was similar to Rrs(555).

### 4.1. Comparison of Rrs

Statistical results for GOCI-derived Rrs values from the four AC methods versus in situ data are given in Tables 1–4, and corresponding scatterplots are shown in Figure 4. The method providing the least accurate Rrs is MUMM, which has the highest uncertainties for all bands. The NASA-STD method performs slightly better than the MUMM algorithm, as reflected in the improved APD and the RMSE ranging from 27.58%–94.55% and 0.0003–0.0028 $sr^{-1}$, respectively, in the VIS bands. The Kd-based algorithm is similar to the KOSC-STD method. Satisfactory results are obtained for the KOSC-STD algorithm, which shows the most accurate performance for a time-window of ±3 h with values of APD and Bias of 22.08%–73.95% and 0.0008–0.0019 $sr^{-1}$, respectively (for a time-window of ±1 h: APD is 12.37%–100.53% and Bias is $-2 \times 10^{-6}$ to $1 \times 10^{-3}$ $sr^{-1}$), and slopes closer to the 1:1 relationship. Overall, the spectrum shapes and the magnitudes at all sites for the KOSC-STD algorithm are much more consistent with in situ Rrs (not shown here).

**Table 1.** Statistical results for GOCI-retrieved Rrs values obtained using the NASA-STD algorithm and the in situ Rrs comparison at 02:16, 03:16, and 04:16 UTC for ±3 h (values in parentheses indicate results for ±1 h). Percentages of negative retrievals (%) are given; these values were removed in the statistics. The italic numerals of each index represent ±3 h, while the statistical results in parentheses are ±1 h. APD: absolute percentage difference, RMSE: Root Mean Square Error.

| Method | Rrs($\lambda$) | %Negative | Count (N) | APD(%) | RMSE($sr^{-1}$) | Bias($sr^{-1}$) | $R^2$ | Slope | Intercept |
|---|---|---|---|---|---|---|---|---|---|
| | Rrs(412) | *11* | *27* | *94.55* | *0.003* | *0.002* | *0.37* | *0.27* | *$1 \times 10^{-3}$* |
| | | (25) | (8) | (63.73) | (0.002) | (0.002) | (0.52) | (0.36) | ($1 \times 10^{-3}$) |
| | Rrs(443) | *7* | *28* | *47.94* | *0.002* | *0.001* | *0.39* | *0.49* | *$1 \times 10^{-3}$* |
| | | (12) | (8) | (38.21) | (0.001) | (0.001) | (0.38) | (0.49) | ($1 \times 10^{-3}$) |
| | Rrs(490) | *0* | *30* | *27.58* | *0.001* | *$8 \times 10^{-4}$* | *0.67* | *0.85* | *$-6 \times 10^{-5}$* |
| | | (0) | (10) | (20.93) | ($9 \times 10^{-4}$) | ($6 \times 10^{-4}$) | (0.57) | (**0.99**) | ($-6 \times 10^{-4}$) |
| NASA-STD | Rrs(555) | *0* | *30* | *29.25* | *0.001* | *$2 \times 10^{-4}$* | *0.60* | *1.16* | *$-8 \times 10^{-4}$* |
| | | (0) | (10) | (**16.64**) | ($6 \times 10^{-4}$) | ($2 \times 10^{-4}$) | (**0.85**) | (0.79) | ($5 \times 10^{-4}$) |
| | Rrs(660) | *0* | *29* | *74.06* | *$4 \times 10^{-4}$* | *$2 \times 10^{-4}$* | *0.56* | *0.55* | *$1 \times 10^{-4}$* |
| | | (0) | (9) | (71.18) | ($3 \times 10^{-4}$) | ($2 \times 10^{-4}$) | (0.31) | (0.36) | ($2 \times 10^{-4}$) |
| | Rrs(680) | *0* | *28* | *79.42* | *$5 \times 10^{-4}$* | *$2 \times 10^{-4}$* | *0.58* | *0.48* | *$1 \times 10^{-4}$* |
| | | (0) | (8) | (89.46) | ($4 \times 10^{-4}$) | ($3 \times 10^{-4}$) | (0.12) | (0.28) | ($3 \times 10^{-4}$) |
| | Rrs(745) | *6* | *29* | *128.39* | *$2 \times 10^{-4}$* | *$-4 \times 10^{-5}$* | *0.03* | *0.19* | *$2 \times 10^{-4}$* |
| | | (11) | (9) | (149.75) | ($2 \times 10^{-4}$) | ($1 \times 10^{-5}$) | (0.49) | (0.31) | ($9 \times 10^{-5}$) |
| | Rrs(865) | *6* | *29* | *196.78* | *$2 \times 10^{-4}$* | *$-1 \times 10^{-5}$* | *0.06* | *0.44* | *$1 \times 10^{-4}$* |
| | | (11) | (9) | (206.72) | (**$9 \times 10^{-5}$**) | (**$4 \times 10^{-7}$**) | (0.29) | (0.41) | ($7 \times 10^{-5}$) |

**Table 2.** Statistical results for GOCI-retrieved Rrs values obtained using the KOSC-STD algorithm and the in situ Rrs comparison at 02:16, 03:16, and 04:16 UTC for ±3 h (values in parentheses indicate results for ±1 h). The presentation style is the same as described in Table 1.

| Method | Rrs($\lambda$) | %Negative | Count (N) | APD(%) | RMSE($sr^{-1}$) | Bias($sr^{-1}$) | $R^2$ | Slope | Intercept |
|---|---|---|---|---|---|---|---|---|---|
| | Rrs(412) | *3* | *28* | *73.95* | *0.002* | *0.002* | *0.43* | *0.38* | *$1 \times 10^{-3}$* |
| | | (11) | (9) | (57.00) | ($2 \times 10^{-3}$) | ($1 \times 10^{-3}$) | (0.64) | (0.42) | ($1 \times 10^{-3}$) |
| | Rrs(443) | *3* | *28* | *35.27* | *0.001* | *0.001* | *0.41* | *0.45* | *$1 \times 10^{-3}$* |
| | | (11) | (9) | (27.33) | ($8 \times 10^{-4}$) | ($6 \times 10^{-4}$) | (0.60) | (0.57) | ($1 \times 10^{-3}$) |
| | Rrs(490) | *0* | *29* | *22.08* | *$9 \times 10^{-4}$* | *$5 \times 10^{-4}$* | *0.49* | *0.66* | *$1 \times 10^{-3}$* |
| | | (0) | (10) | (17.16) | ($7 \times 10^{-4}$) | ($4 \times 10^{-4}$) | (0.75) | (0.74) | ($8 \times 10^{-4}$) |
| KOSC-STD | Rrs(555) | *0* | *29* | *22.23* | *$7 \times 10^{-4}$* | *$1 \times 10^{-4}$* | *0.66* | *0.78* | *$5 \times 10^{-4}$* |
| | | (0) | (10) | (**12.37**) | ($4 \times 10^{-4}$) | ($1 \times 10^{-5}$) | (**0.90**) | (0.78) | ($7 \times 10^{-4}$) |
| | Rrs(660) | *0* | *28* | *69.79* | *$3 \times 10^{-4}$* | *$1 \times 10^{-4}$* | *0.14* | *0.30* | *$2 \times 10^{-4}$* |
| | | (0) | (10) | (61.63) | ($3 \times 10^{-4}$) | ($9 \times 10^{-5}$) | (0.08) | (0.27) | ($3 \times 10^{-4}$) |
| | Rrs(680) | *0* | *28* | *68.92* | *$3 \times 10^{-4}$* | *$2 \times 10^{-4}$* | *0.23* | *0.39* | *$2 \times 10^{-4}$* |
| | | (0) | (10) | (100.53) | ($4 \times 10^{-4}$) | ($2 \times 10^{-4}$) | ($5 \times 10^{-3}$) | (0.11) | ($4 \times 10^{-4}$) |
| | Rrs(745) | *0* | *29* | *103.27* | *$1 \times 10^{-4}$* | *$4 \times 10^{-6}$* | *0.72* | *0.99* | *$1 \times 10^{-5}$* |
| | | (0) | (10) | (95.65) | (**$7 \times 10^{-5}$**) | ($-3 \times 10^{-5}$) | (0.66) | (0.87) | ($7 \times 10^{-5}$) |
| | Rrs(865) | *0* | *29* | *140.56* | *$1 \times 10^{-4}$* | *$4 \times 10^{-6}$* | *0.60* | *1.34* | *$2 \times 10^{-5}$* |
| | | (0) | (10) | (125.78) | (**$7 \times 10^{-5}$**) | (**$-2 \times 10^{-6}$**) | (0.53) | (0.84) | ($4 \times 10^{-5}$) |

**Table 3.** Statistical results for GOCI-retrieved Rrs values obtained using the Kd-based algorithm and the in situ Rrs comparison at 02:16, 03:16, and 04:16 UTC for ±3 h (values in parentheses indicate results for ±1 h). The presentation style is the same as described in Table 1.

| Method | Rrs(λ) | %Negative | Count (N) | APD(%) | RMSE(sr⁻¹) | Bias(sr⁻¹) | R² | Slope | Intercept |
|---|---|---|---|---|---|---|---|---|---|
| Kd-based | Rrs(412) | 7 | 29 | 53.64 | 0.002 | 0.001 | 0.56 | 0.41 | $1 \times 10^{-3}$ |
| | | (11) | (9) | (56.80) | (0.002) | (0.001) | (0.60) | (0.32) | **(0.001)** |
| | Rrs(443) | 7 | 29 | 32.07 | 0.001 | $8 \times 10^{-4}$ | 0.45 | 0.51 | $1 \times 10^{-3}$ |
| | | (11) | (9) | (28.61) | (0.001) | $(8 \times 10^{-4})$ | (0.66) | (0.52) | **(0.001)** |
| | Rrs(490) | **0** | **31** | 22.31 | 0.001 | $6 \times 10^{-4}$ | 0.42 | 0.69 | $6 \times 10^{-4}$ |
| | | (0) | (10) | (17.44) | $(8 \times 10^{-4})$ | $(6 \times 10^{-4})$ | (0.63) | (0.86) | $(8 \times 10^{-5})$ |
| | Rrs(555) | **0** | **31** | 23.68 | $8 \times 10^{-4}$ | $4 \times 10^{-4}$ | 0.65 | 0.36 | $2 \times 10^{-4}$ |
| | | (0) | (10) | **(14.63)** | $(6 \times 10^{-4})$ | $(3 \times 10^{-4})$ | **(0.91)** | (0.77) | $(5 \times 10^{-4})$ |
| | Rrs(660) | **0** | **31** | 72.72 | $3 \times 10^{-4}$ | $2 \times 10^{-4}$ | 0.35 | 0.34 | $2 \times 10^{-4}$ |
| | | (0) | (10) | (71.12) | $(4 \times 10^{-4})$ | $(2 \times 10^{-4})$ | (0.28) | (0.34) | $(2 \times 10^{-4})$ |
| | Rrs(680) | **0** | **31** | 72.76 | $4 \times 10^{-4}$ | $2 \times 10^{-4}$ | 0.34 | 0.34 | $2 \times 10^{-4}$ |
| | | (0) | (10) | (85.49) | $(5 \times 10^{-4})$ | $(3 \times 10^{-4})$ | (0.39) | (0.36) | $(2 \times 10^{-4})$ |
| | Rrs(745) | 3 | 30 | 117.97 | $9 \times 10^{-5}$ | $5 \times 10^{-5}$ | 0.67 | 0.67 | $2 \times 10^{-5}$ |
| | | (11) | (9) | (62.91) | $(7 \times 10^{-5})$ | $(4 \times 10^{-5})$ | (0.75) | (0.63) | $(2 \times 10^{-5})$ |
| | Rrs(865) | 3 | 30 | 170.03 | $1 \times 10^{-4}$ | $4 \times 10^{-5}$ | 0.64 | 0.84 | $1 \times 10^{-5}$ |
| | | (11) | (9) | (138.48) | $\mathbf{(6 \times 10^{-5})}$ | $\mathbf{(3 \times 10^{-5})}$ | (0.81) | **(0.88)** | $(2 \times 10^{-5})$ |

**Table 4.** Statistical results for GOCI-retrieved Rrs values obtained using the MUMM algorithm and the in situ Rrs comparison at 02:16, 03:16, and 04:16 UTC for ±3 h (values in parentheses indicate results for ±1 h). The presentation style is the same as described in Table 1.

| Method | Rrs(λ) | %Negative | Count (N) | APD(%) | RMSE(sr⁻¹) | Bias(sr⁻¹) | R² | Slope | Intercept |
|---|---|---|---|---|---|---|---|---|---|
| MUMM | Rrs(412) | 13 | 31 | 107.47 | 0.003 | 0.002 | 0.10 | 0.14 | $2 \times 10^{-3}$ |
| | | (20) | (10) | (74.20) | (0.002) | (0.002) | (0.44) | (0.23) | (0.002) |
| | Rrs(443) | 6 | 31 | 60.41 | 0.002 | 0.002 | 0.28 | 0.33 | $2 \times 10^{-3}$ |
| | | (20) | (10) | (46.03) | (0.002) | (0.001) | (0.24) | (0.32) | $(2 \times 10^{-3})$ |
| | Rrs(490) | **0** | **32** | 30.87 | 0.001 | $7 \times 10^{-4}$ | 0.32 | 0.59 | $\mathbf{1 \times 10^{-3}}$ |
| | | (0) | (11) | (27.14) | (0.001) | $(7 \times 10^{-4})$ | (0.36) | (0.73) | $(6 \times 10^{-4})$ |
| | Rrs(555) | **0** | **32** | 38.33 | 0.001 | $5 \times 10^{-4}$ | 0.46 | **0.93** | $-4 \times 10^{-4}$ |
| | | (0) | (11) | **(24.18)** | $(9 \times 10^{-4})$ | $(6 \times 10^{-4})$ | **(0.80)** | (0.73) | $(4 \times 10^{-4})$ |
| | Rrs(660) | **0** | 31 | 97.20 | 0.001 | $3 \times 10^{-4}$ | 0.71 | 1.53 | $-6 \times 10^{-4}$ |
| | | (0) | (11) | (93.42) | $(5 \times 10^{-4})$ | $(3 \times 10^{-4})$ | (0.30) | (0.22) | $(3 \times 10^{-4})$ |
| | Rrs(680) | **0** | **32** | 101.78 | $6 \times 10^{-4}$ | $3 \times 10^{-4}$ | 0.76 | 0.69 | $-2 \times 10^{-4}$ |
| | | (0) | (10) | (125.82) | $(6 \times 10^{-4})$ | $(4 \times 10^{-4})$ | (0.22) | (0.22) | $(3 \times 10^{-4})$ |
| | Rrs(745) | 6 | **32** | 142.57 | $2 \times 10^{-4}$ | $4 \times 10^{-5}$ | 0.52 | 0.51 | $8 \times 10^{-5}$ |
| | | (18) | (11) | (120.50) | $(2 \times 10^{-4})$ | $(8 \times 10^{-5})$ | (0.35) | (0.23) | $(7 \times 10^{-5})$ |
| | Rrs(865) | 6 | **32** | 193.32 | $\mathbf{1 \times 10^{-4}}$ | $\mathbf{3 \times 10^{-5}}$ | 0.13 | 0.36 | $9 \times 10^{-5}$ |
| | | (18) | (11) | (244.88) | $(2 \times 10^{-4})$ | $(9 \times 10^{-5})$ | (0.53) | (0.30) | $(3 \times 10^{-5})$ |

The difference in the APD of Rrs(555) between the ±1 and the ±3 h time-windows is <15% for the four algorithms. Hence, scatterplots between the GOCI-retrieved Rrs and the in situ measurements of ±3 h match-up time are presented in Figure 4. Between 412 and 555 nm, all four AC methods overestimate Rrs, especially at the shortest wavelengths (412 and 443 nm). For Rrs(412) and Rrs(443), the Kd-based method performs best. For wavelengths at 745 and 865 nm, all four AC methods underestimate Rrs. The results show good agreement between the GOCI-estimated Rrs and the in situ measured Rrs, especially at 490 and 555 nm, with values of APD of 22.08%–30.87% and 22.23%–38.33%, respectively. The highest accuracy is observed for Rrs(555). The largest APD errors are observed at 745 and 865 nm (103.27%–196.78%) followed by 412 nm (73.95%–107.47%).

The difference between the GOCI-derived and the field-based Rrs data as a function of wavelength among the three GOCI overpass times is shown in Figure 5. The APD at 04:16 UTC is lower than at 02:16 and 03:16 UTC. The retrievals at Rrs(555) are the most accurate for the three overpass times for all four AC methods. The values of RMSE and Bias generally decrease with increasing wavelength. The lowest RMSE at 04:16 UTC occurs with the KOSC-STD method. Bias decreases quickly as wavelength increases from 412 to 490 nm, although it is nearly stable at wavelengths beyond 490 nm for all overpass times; this is especially true at 04:16 UTC.

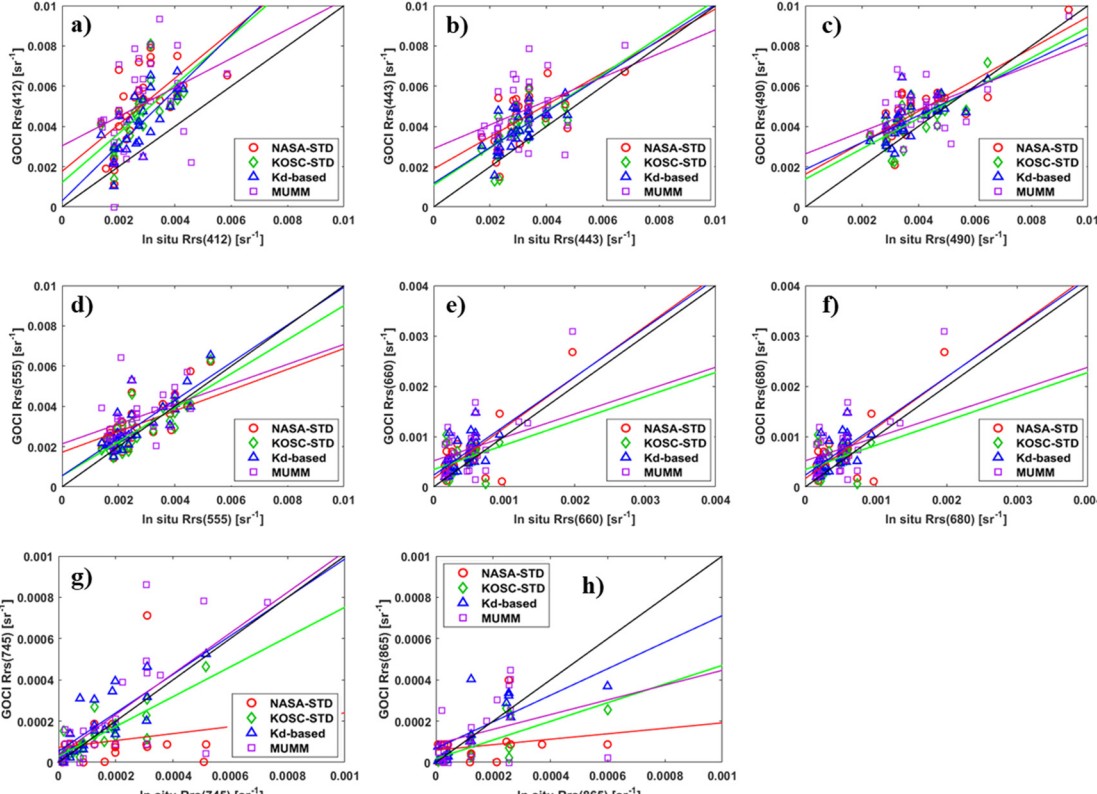

**Figure 4.** Scatterplots of GOCI Rrs retrieved using the four algorithms versus the in situ Rrs (**a–h**) at 02:16, 03:16, and 04:16 UTC within a ±3 h match-up time for eight GOCI wavelengths. Red circles, green diamonds, blue triangles, and purple squares represent scatter points of NASA-STD, KOSC-STD, Kd-based, and MUMM algorithms, respectively, and red, green, blue, and purple solid lines are the corresponding fitted lines. The black solid line represents the 1:1 line.

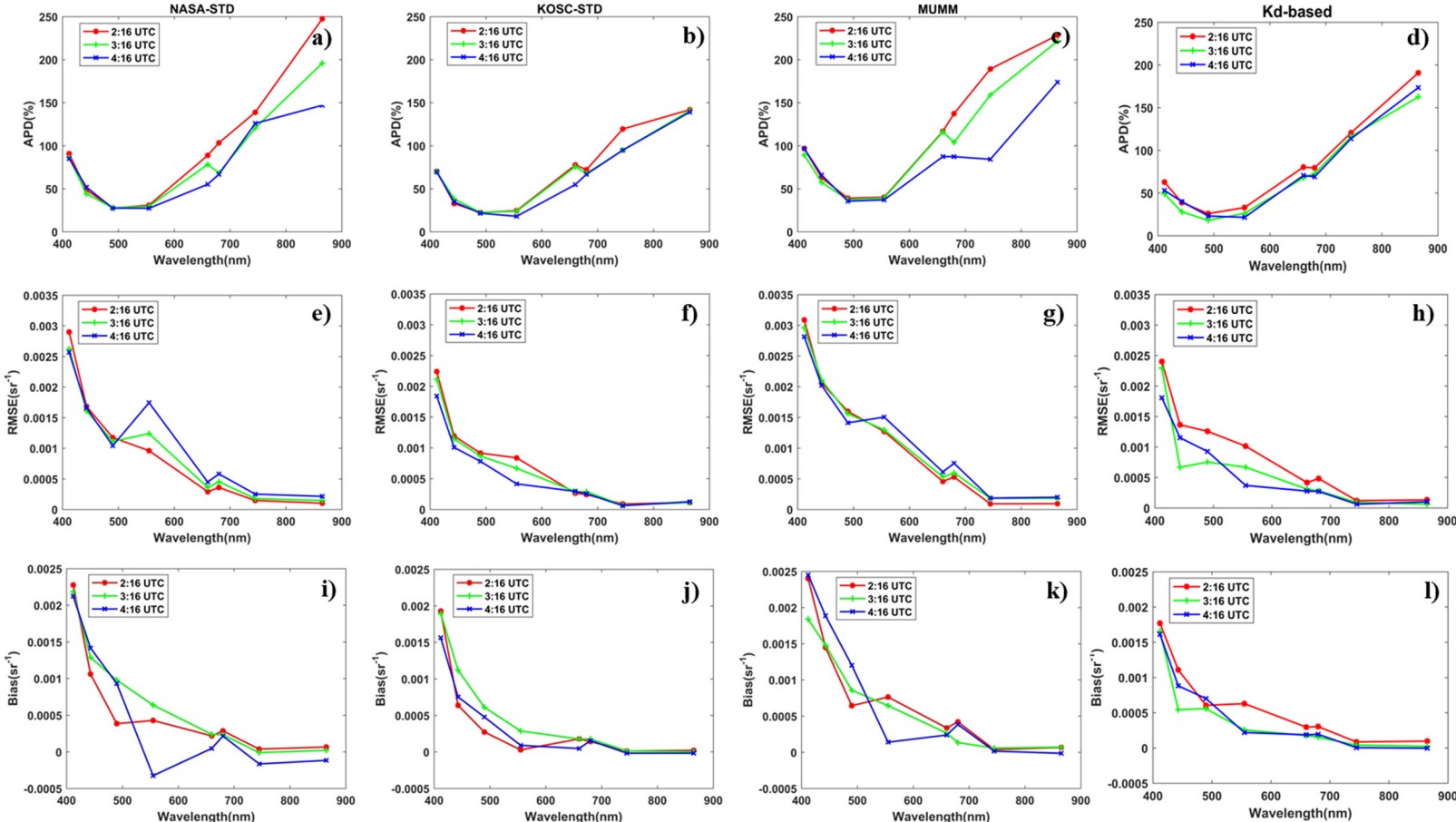

**Figure 5.** APD (%, upper row), RMSE (sr$^{-1}$, middle row), and Bias (sr$^{-1}$, lower row) of Rrs per period (02:16, 03:16, 04:16 UTC) and per algorithm (from left to right column: NASA-STD, KOSC-STD, MUMM, and Kd-based methods) as a function of wavelength at ±3 h. Red, green, and blue solid lines represent the GOCI scenes at 02:16, 03:16, 04:16 UTC, respectively.

To develop bio-optical algorithms, band ratios of remote-sensing reflectance, i.e., Rrs(443)/Rrs(555) and Rrs(490)/Rrs(555), are commonly used to retrieve biogeochemical parameters [46,47]. The band ratios for each algorithm were evaluated against in situ band ratios (Table 5 and Figure 6). The distribution of the ratio Rrs(490)/Rrs(555) is much more concentrated than Rrs(443)/Rrs(555), and the former is closer to the 1:1 line (Figure 6). Within a ±3 h match-up time, the ratio Rrs(490)/Rrs(555) exhibits a relatively better performance (APD: 11%–19%, RMSE: 0.272–0.309 sr$^{-1}$) in comparison with Rrs(443)/Rrs(555) for all four methods. The ratio Rrs(490)/Rrs(555) retrieved by the Kd-based algorithm has the best consistency with the in situ data, whereas Rrs(443)/Rrs(555) has the poorest performance for the KOSC-STD algorithm.

**Table 5.** Statistical results for the retrieved values of two band ratios, Rrs(443)/Rrs(555) and Rrs(490)/Rrs(555), obtained using the NASA-STD, the KOSC-STD, the Kd-based, and the MUMM algorithms. The italic numerals of each index represent ±3 h, while the statistical results in parentheses are ±1 h.

| Method | Band Ratio | APD(%) | RMSE(sr$^{-1}$) | Bias(sr$^{-1}$) | R$^2$ | Slope | Intercept |
|---|---|---|---|---|---|---|---|
| NASA-STD | Rrs(443)/Rrs(555) | *34.11* (34.14) | *0.424* (0.414) | *0.287* (0.326) | *0.61* (**0.74**) | *0.60* (0.59) | *0.29* (0.24) |
| | Rrs(490)/Rrs(555) | *18.73* (**14.73**) | *0.309* (**0.252**) | ***0.089*** (0.145) | *0.40* (0.72) | *0.57* (**0.65**) | ***0.57*** (0.37) |
| KOSC-STD | Rrs(443)/Rrs(555) | *27.05* (29.27) | *0.402* (0.428) | *0.257* (0.282) | *0.63* (**0.80**) | ***0.66*** (0.50) | *0.26* (0.38) |
| | Rrs(490)/Rrs(555) | ***14.49*** (19.45) | *0.309* (0.414) | *0.119* (0.224) | *0.58* (0.79) | *0.57* (0.45) | *0.59* (0.63) |
| Kd-based | Rrs(443)/Rrs(555) | *23.01* (19.61) | *0.332* (0.306) | *0.149* (0.160) | *0.46* (**0.72**) | ***0.64*** (0.58) | *0.34* (0.35) |
| | Rrs(490)/Rrs(555) | ***11.43*** (14.55) | *0.272* (0.386) | ***0.054*** (0.139) | *0.49* (0.66) | *0.55* (0.44) | *0.64* (**0.69**) |
| MUMM | Rrs(443)/Rrs(555) | *28.19* (26.61) | *0.338* (0.313) | *0.194* (0.190) | *0.54* (**0.56**) | ***0.81*** (0.60) | *0.07* (0.33) |
| | Rrs(490)/Rrs(555) | ***14.92*** (16.56) | *0.287* (**0.263**) | *−0.054* (**0.012**) | *0.32* (0.37) | *0.57* (0.53) | ***0.67*** (0.65) |

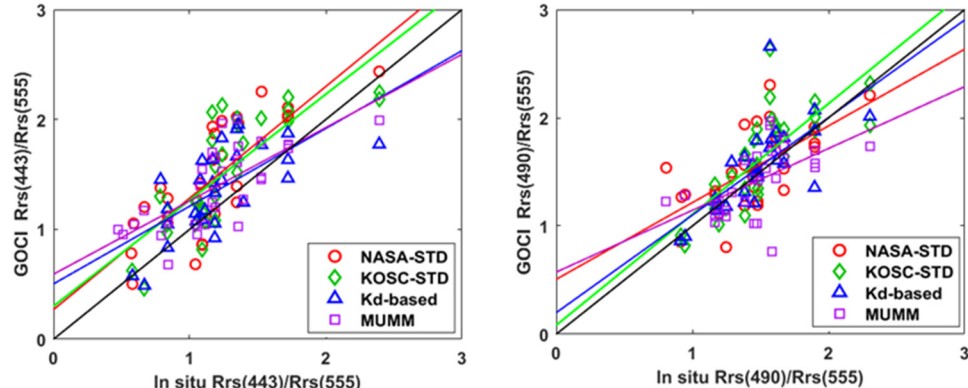

**Figure 6.** Scatterplots of band ratios Rrs(443)/Rrs(555) and Rrs(490)/Rrs(555) between the GOCI-retrieved and the in situ ratios at 02:16, 03:16, and 04:16 UTC within a ±3 h match-up time-window. Symbols are the same as described in Figure 4.

In conclusion, the ratio Rrs(490)/Rrs(555) estimated by all four AC methods results in better retrievals, which means this band ratio is more suitable for retrieving the biogeochemical properties of the YS region in summer. In comparison with the three single band values, i.e., Rrs(443), Rrs(490), and Rrs(555) (Tables 1–4), the band ratios reduce the systematic uncertainty of the AC procedure. Furthermore, the band ratios also improve the strength of the correlations relative to the single bands.

## 4.2. Comparison of AOT and AE

Aerosol optical products are by-products derived from the AC algorithms. To understand the uncertainties of aerosol products for AC schemes, it is necessary to analyze the features of aerosols. Many studies have investigated the wavelength dependence of AOT and the AE. The accuracy of the retrievals of AOT(443), AOT(555), AOT(680), AOT(865), and AE(443,865) of our current study is presented in Tables 6–9 and Figure 7. Summary statistics for GOCI versus field data aerosol products of ±3 and ±1 h are also shown in Table 3. Within a ±3 h match-up time for AOT, the MUMM algorithm is the least accurate method in the visible bands (APD: 44.60%–66.12%, RMSE: 0.095–0.151 sr$^{-1}$, Bias: −0.016 to −0.080 sr$^{-1}$). Conversely, the KOSC-STD algorithm is the most accurate, providing the best estimations of AOT (APD: 39.83%–75.26%, RMSE: 0.092–0.147 sr$^{-1}$, Bias: −0.003 to −0.051 sr$^{-1}$). The Kd-based NIR correction algorithm produces values of APD of 40.89%–71.77%, RMSE of 0.094–0.148 sr$^{-1}$, and Bias of −0.005 to 0.036 sr$^{-1}$, which are slightly worse than the KOSC-STD method. None of the algorithms are able to estimate AE(443,865) precisely, and low correlations are found for NASA-STD (0.11), KOSC-STD (0.13), Kd-based (0.09), and MUMM (0.12) algorithms. Within a ±1 h match-up time, except for the KOSC-STD and the Kd-based algorithms for AOT(443), other parameters are estimated better than with a ±3 h time-window. However, the number of match-ups is around 30 for the ±3 h time-window for each algorithm, whereas it is only 10 for the ±1 h time-window. Therefore, in this study, the temporal window of the study was extended to ±3 h around the GOCI overpass time (at 02:16, 03:16, and 04:16 UTC, separately) to extend the number of potential match-ups.

**Table 6.** Validation statistics for the NASA-STD method of GOCI-derived AOT(443), AOT(555), AOT(680), AOT(865), and AE(443,865) at 02:16, 03:16, and 04:16 UTC: APD (%), RMSE (sr$^{-1}$), Bias (sr$^{-1}$), $R^2$ (dimensionless), Slope (dimensionless), Intercept (dimensionless). Count (N) indicates the total number of match-ups. The italic numerals of each index represent ±3 h, while the statistical results in parentheses are ±1 h.

| Method | Parameter | Count (N) | APD(%) | RMSE(sr$^{-1}$) | Bias(sr$^{-1}$) | $R^2$ | Slope | Intercept |
|---|---|---|---|---|---|---|---|---|
| NASA-STD | AOT(443) | *27* (8) | *41.29* (38.38) | *0.148* (0.090) | *−0.083* (−0.082) | *0.41* **(0.70)** | *0.78* **(1.14)** | *0.12* (0.06) |
| | AOT(555) | **30** (10) | *40.88* (33.81) | *0.106* (0.083) | *−0.063* (−0.067) | *0.57* (0.49) | *0.91* (1.13) | *0.08* (0.05) |
| | AOT(680) | *28* (8) | *42.46* **(30.23)** | *0.099* (0.069) | *−0.031* (−0.038) | *0.37* (0.38) | *0.62* (1.07) | *0.09* (0.03) |
| | AOT(865) | *29* (9) | *78.99* (37.72) | *0.104* **(0.066)** | *0.011* **(−0.003)** | *0.17* (0.32) | *0.47* (1.10) | *0.07* (9 × 10$^{-3}$) |
| | AE(443,865) | *27* (8) | *209.81* (138.54) | *0.808* (0.546) | *−0.926* (−0.425) | *0.11* (0.61) | *0.30* (1.13) | *0.45* (0.35) |

**Table 7.** Validation statistics for the KOSC-STD method of GOCI-derived AOT(443), AOT(555), AOT(680), AOT(865), and AE(443,865) at 02:16, 03:16, and 04:16 UTC: APD (%), RMSE (sr$^{-1}$), Bias (sr$^{-1}$), $R^2$ (dimensionless), Slope (dimensionless), Intercept (dimensionless). The presentation style is the same as described in Table 6.

| Method | Parameter | Count (N) | APD(%) | RMSE(sr$^{-1}$) | Bias(sr$^{-1}$) | $R^2$ | Slope | Intercept |
|---|---|---|---|---|---|---|---|---|
| KOSC-STD | AOT(443) | 28 (9) | *40.51* (50.66) | *0.148* (0.161) | *−0.051* (0.019) | *0.46* (0.19) | *0.70* (0.17) | ***0.12*** (0.18) |
| | AOT(555) | 29 (10) | ***39.83*** (22.79) | ***0.092*** (0.051) | *−0.040* (−0.025) | ***0.69*** (0.65) | ***0.82*** (0.74) | *0.08* (0.07) |
| | AOT(680) | 28 (10) | *41.25* (25.48) | *0.098* (0.054) | ***−0.004*** (9 × 10$^{-4}$) | *0.55* (0.53) | *0.60* (0.74) | *0.08* (0.04) |
| | AOT(865) | 29 (10) | *75.26* (64.42) | *0.135* (0.091) | *0.035* (0.021) | *0.25* (0.08) | *0.35* (0.34) | *0.09* (0.07) |
| | AE(443,865) | 28 (9) | *206.82* (262.02) | *0.760* (0.672) | *−0.842* (−0.439) | *0.13* (0.20) | *0.15* (0.62) | *0.08* (0.65) |

**Table 8.** Validation statistics for the Kd-based method of GOCI-derived AOT(443), AOT(555), AOT(680), AOT(865), and AE(443,865) at 02:16, 03:16, and 04:16 UTC: APD (%), RMSE (sr$^{-1}$), Bias (sr$^{-1}$), R$^2$ (dimensionless), Slope (dimensionless), Intercept (dimensionless). The presentation style is the same as described in Table 6.

| Method | Parameter | Count (N) | APD(%) | RMSE(sr$^{-1}$) | Bias(sr$^{-1}$) | R$^2$ | Slope | Intercept |
|---|---|---|---|---|---|---|---|---|
| Kd-based | AOT(443) | *29* (9) | ***40.89*** (51.82) | *0.148* (0.162) | *−0.051* (0.016) | *0.46* (0.17) | *0.70* (0.16) | *0.12* (0.18) |
| | AOT(555) | *31* (10) | *41.17* (33.53) | ***0.094*** (0.066) | *−0.039* (−0.030) | ***0.70*** (0.59) | ***0.78*** (0.57) | *0.08* (0.10) |
| | AOT(680) | *31* (10) | *42.18* (28.61) | *0.097* (0.056) | ***−0.005*** (−0.005) | *0.57* (0.50) | *0.60* (0.71) | *0.08* (0.05) |
| | AOT(865) | *30* (10) | *71.77* (57.95) | *0.131* (0.079) | *0.036* (0.024) | *0.29* (0.25) | *0.38* (0.58) | *0.08* (0.04) |
| | AE(443,865) | *29* (9) | *202.36* (275.17) | *0.551* (0.616) | *−0.231* (−0.262) | *0.09* (0.17) | *0.71* (0.47) | *0.44* (0.66) |

**Table 9.** Validation statistics for the MUMM method of GOCI-derived AOT(443), AOT(555), AOT(680), AOT(865), and AE(443,865) at 02:16, 03:16, and 04:16 UTC: APD (%), RMSE (sr$^{-1}$), Bias (sr$^{-1}$), R$^2$ (dimensionless), Slope (dimensionless), Intercept (dimensionless). The presentation style is the same as described in Table 6.

| Method | Parameter | Count (N) | APD(%) | RMSE(sr$^{-1}$) | Bias(sr$^{-1}$) | R$^2$ | Slope | Intercept |
|---|---|---|---|---|---|---|---|---|
| MUMM | AOT(443) | *29* (9) | ***40.89*** (51.82) | *0.148* (0.162) | *−0.051* (0.016) | *0.46* (0.17) | *0.70* (0.16) | *0.12* (0.18) |
| | AOT(555) | *31* (10) | *41.17* (33.53) | ***0.094*** (0.066) | *−0.039* (−0.030) | ***0.70*** (0.59) | ***0.78*** (0.57) | *0.08* (0.10) |
| | AOT(680) | *31* (10) | *42.18* (28.61) | *0.097* (0.056) | ***−0.005*** (−0.005) | *0.57* (0.50) | *0.60* (0.71) | *0.08* (0.05) |
| | AOT(865) | *30* (10) | *71.77* (57.95) | *0.131* (0.079) | *0.036* (0.024) | *0.29* (0.25) | *0.38* (0.58) | *0.08* (0.04) |
| | AE(443,865) | *29* (9) | *202.36* (275.17) | *0.551* (0.616) | *−0.231* (−0.262) | *0.09* (0.17) | *0.71* (0.47) | *0.44* (0.66) |

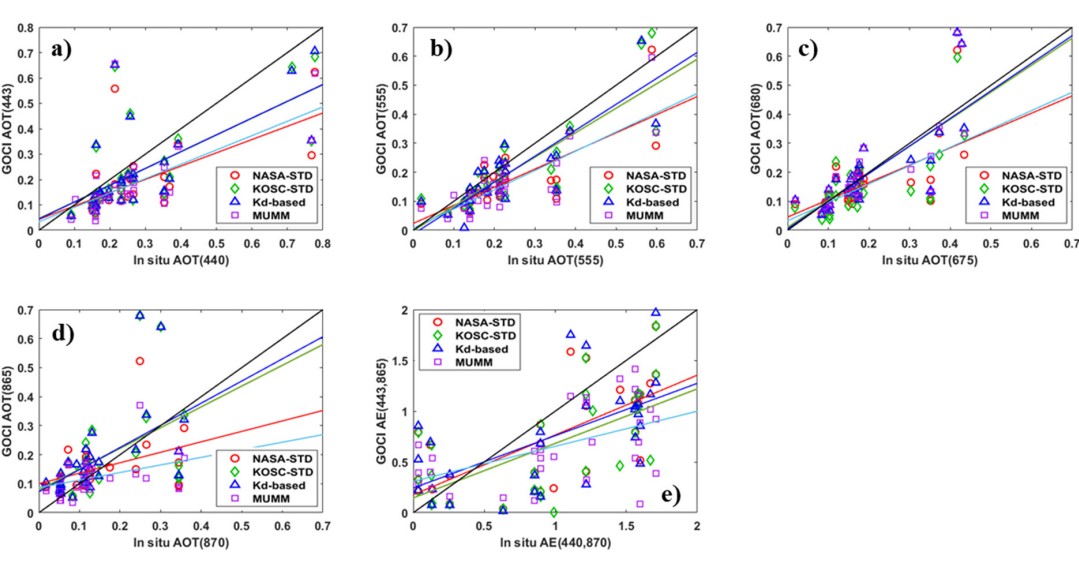

**Figure 7.** Scatterplots of GOCI-estimated (**a**) AOT(443), (**b**) AOT(555), (**c**) AOT(675), (**d**) AOT(865), and (**e**) AE(443,865) versus in situ aerosol products within a ± 3 h match-up time. Symbols are the same as described in Figure 4.

Figure 7 displays scatterplots of ±3 h windows between the GOCI-retrieved AOT versus the in situ AOT at four GOCI bands and AE(443,865) at 02:16, 03:16, and 04:16 UTC. The results of the AOT data show that the four methods exhibit underestimation at AOT bands. AOT(443) and AOT(865) retrievals present the highest RMSEs and relative errors, while AOT(555) performs best in the YS. All retrieved AE values from the four algorithms are <2.0, with the majority in the range 1.2–1.6. This indicates a size distribution dominated by fine particle sizes, which are considered absorbing aerosol types (coastal and pollution aerosols) [34,61–63]. The distribution of the retrieved values around the 1:1 line are more scattered for AE(443,865) than AOT, which is consistent with previous published articles [13,14].

To analyze the error between the retrieved and the in situ AOT (at 443, 555, 680, and 865 nm) at the three GOCI overpass times, Figure 8 illustrates the APD, the RMSE, and the Bias per time and per method between the satellite retrieval values and the field-based AOT. The spectral behavior of APD for AOT is similar to that of Rrs (Figure 5, upper row). The performance of each algorithm is best at 04:16 UTC and worst at 02:16 UTC. This is consistent with the results obtained for Rrs. For all approaches, the APD in the visible bands is stable over the spectrum with a peak at 865 nm (around 50% at 04:16 UTC and near 100% at 02:16 UTC, Figure 8a–d). The sign of the bias is the same for the four AC methods, i.e., negative values in blue bands and positive values in red and NIR bands. The time of image acquisition affects the magnitude of the errors but not the shape. The aerosol optical properties generated from the four methods can be generalized as follows: (1) the MUMM algorithm produces the largest uncertainties in retrievals for both AOT and AE(443,865), the NASA-STD performs slightly better than MUMM, the performance of KOSC-STD is similar to the Kd-based algorithm, and the Kd-based algorithm obtains the most accurate values; (2) for all AC methods, the values of AOT at the four GOCI bands and AE(443,865) are underestimated.

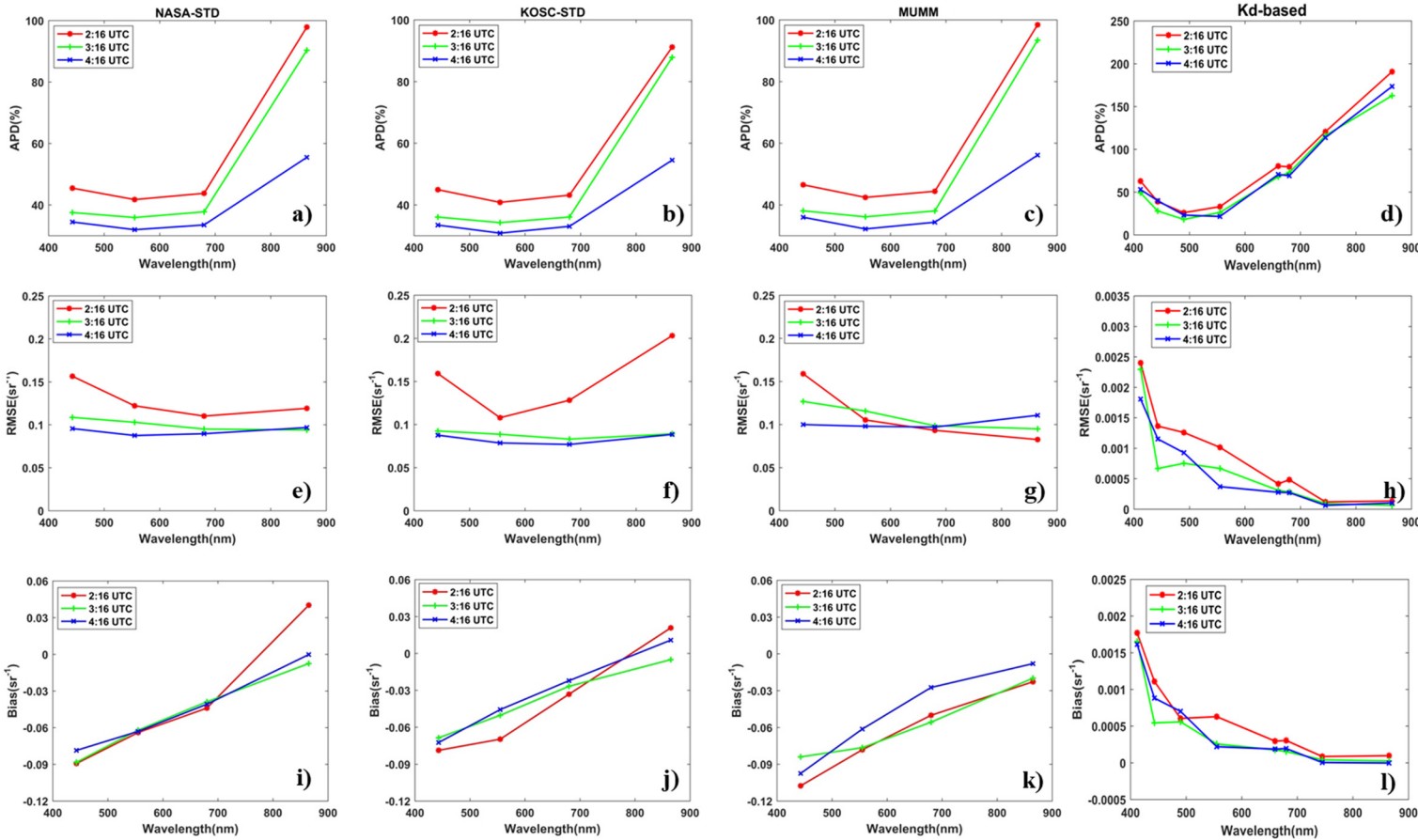

**Figure 8.** APD (%, upper row), RMSE (sr$^{-1}$, middle row), and Bias (sr$^{-1}$, lower row) of AOT per period (02:16, 03:16, and 04:16 UTC) and per algorithm (from left to right column: NASA-STD, KOSC-STD, MUMM, and Kd-based methods) as a function of wavelength. Symbols are the same as described in Figure 5. The four points on each line in each panel represent AOT at four wavelengths: 443, 555, 680, and 865 nm.

### 4.3. Comparison of OC3G Chla Retrievals

GOCI-OC3G Chla retrievals from the four AC methods versus in situ Chla concentrations were compared for the two different match-up windows (±1 and ±3 h) (Table 10). Within a ±1 h match-up window, all algorithms show values of APD of 22.05%–48.73%, RMSE of 0.141–0.215 mg m$^{-3}$, and Bias of 0.010–0.092 mg m$^{-3}$. Within a ±3 h match-up window, all methods show values of APD of 33.47%–48.73, RMSE of 0.146–0.244 mg m$^{-3}$, and Bias of 0.043–0.102 mg m$^{-3}$. There is a slight difference between the two time-windows regarding the estimation of Chla. Of the four methods, the Kd-based algorithm exhibits the best Chla retrievals at both ±3 and ±1 h windows, with 95% confidence interval of (0.36, 0.63) (±3 h window) and (0.27, 0.94) (±1 h window). The APD values of Chla derived using the Kd-based algorithm range from 22.05% (±1 h window) to 33.47% (±3 h window). The KOSC-STD algorithm exhibits similar results to the Kd-based Chla retrievals with APD values of 30.11% (±1 h window) to 37.56% (±3 h window). The MUMM method shows the lowest uncertainty, RMSE, and Bias for the ±3 h window.

**Table 10.** Statistical results for the retrieved values of chlorophyll-a (Chla) obtained using NASA-STD, KOSC-STD, Kd-based, and MUMM algorithms and in situ Rrs comparison at 02:16, 03:16, and 04:16 UTC. The italic numerals of each index represent ±3 h, while the statistical results in parentheses are ±1 h.

| Method | Parameter | Confidence Interval (mg m$^{-3}$) | APD(%) | RMSE(sr$^{-1}$) | Bias(sr$^{-1}$) | R$^2$ | Slope | Intercept |
|---|---|---|---|---|---|---|---|---|
| NASA-STD | Chla | *(0.41,0.66)* | *45.43* | *0.235* | *0.084* | *0.71* | *0.82* | *0.29* |
| | | (0.33,0.84) | (39.82) | (0.215) | (0.068) | (0.76) | (0.92) | (0.21) |
| KOSC-STD | Chla | *(0.35,0.58)* | *37.56* | *0.146* | *0.043* | *0.85* | ***0.84*** | *−0.01* |
| | | (0.27,0.86) | (30.11) | (**0.141**) | (**0.010**) | (**0.92**) | (0.88) | (0.01) |
| NIR-corrected | Chla | *(0.36,0.63)* | *33.47* | *0.160* | *0.063* | *0.83* | *0.76* | *0.01* |
| | | (0.27,0.94) | (**22.05**) | (0.168) | (0.054) | (0.87) | (0.79) | (0.04) |
| MUMM | Chla | *(0.37,0.50)* | *48.73* | *0.244* | *0.102* | *0.46* | *0.64* | ***0.35*** |
| | | (0.27,0.52) | (31.85) | (0.163) | (0.092) | (0.67) | (0.61) | (0.34) |

Figure 9 displays scatterplots of GOCI Chla retrieved by the four AC algorithms versus in situ Chla. The Chla concentrations are mainly <1 mg m$^{-3}$. In this range, Chla estimated by both the KOSC-STD and the Kd-based methods is slightly overestimated, while the values from the NASA-STD and the MUMM algorithms cluster around the 1:1 line.

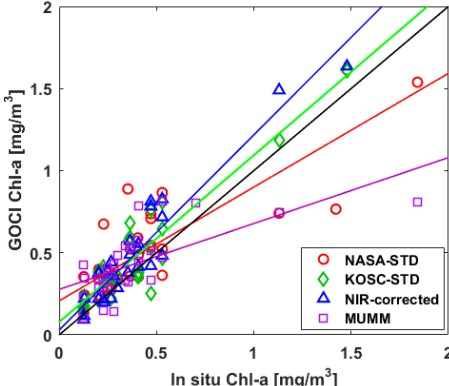

**Figure 9.** Scatterplots of GOCI Chla retrieved by the four algorithms versus in situ Chla at 02:16, 03:16, and 04:16 UTC within a ±3 h match-up time. Symbols are the same as described in Figure 4.

## 5. Discussion

This research compared four AC algorithms (NASA-STD, KOSC-STD, Kd-based, and MUMM) using in situ aerosol optical properties (AOT and AE) as well as above-water Rrs and Chla concentration.

These four algorithms extend the GW-AC algorithm to Case-2 waters for which the black-pixel assumption cannot be considered. The features of GOCI-derived Rrs, aerosol optical products, and Chla concentrations derived using the four algorithms can be summarized as follows. (1) Match-up time-windows of ±1 and ±3 h were compared for AOT, Rrs, and Chla (Tables 1–4) to obtain the best balance in achieving the highest number of match-ups and reducing the water mass and particle dynamics. Overall, all methods had better performance with a ±1 h time-window. However, the number of match-ups was too low, thus the match-up time-window was extended to ±3 h to ensure a suitable number of match-ups (Figures 3–8); (2) all methods overestimated Rrs and underestimated AOT data in comparison with in situ values; (3) for visible bands, all algorithms performed well for Rrs and AOT at 555 nm, but larger uncertainties appeared at other wavelengths (especially 412 nm) primarily because of incorrect estimation of the NIR ocean contributions [4]. For NIR bands, the error is larger than for the visible bands; (4) GOCI-derived Rrs and AOT were sensitive to the image acquisition time. The errors were lower for images acquired at 04:16 UTC in comparison with the two other times (02:16 and 03:16 UTC) because the solar and the satellite angles were lower at 04:16 UTC; (5) for GOCI-OC3G Chla retrievals, the Kd-based and the KOSC-STD algorithms had lower uncertainties for Rrs, AOT, and AE than the other two algorithms. However, the strengths and the weaknesses of each approach affected the accuracy of the AC procedure. The discrepancy could possibly have been caused by uncertainties in the NIR correction models, the aerosol models, or GOCI slot effects.

The essential distinction between the four AC algorithms is the NIR correction model adopted. In our study and with a limited dataset, the KOSC-STD approach appears the method most appropriate for Rrs retrieval, whereas the Kd-based algorithm is slightly better than KOSC-STD for estimation of Chla over the YS region. For the KOSC-STD algorithm, the Lw at 660 nm can be calculated, then the GW-AC algorithm iteratively updates the fields of Lw in the NIR bands. This method has been validated in turbid coastal regions such as the Korean Peninsula [3,27,64]. However, the KOSC-STD algorithm has a further limitation, i.e., the polynomial equation of Lw between the red–NIR bands is variable depending on the concentrations of suspended particulate matter, CDOM, and Chla [62,65]. Similarly, for the Kd-based method, which is a regional model for the GOCI overpass area, Kd(490) plays an important role in the iterative computation of Rrs in the NIR bands. On the one hand, the relationship between Kd(490) and NIR nLw might not fit for turbid waters in the short term. On the other hand, nLw(660) might be insensitive to changes of nLw(745) in extremely turbid waters [66]. In other words, nLw(660) would become constant, meaning that Kd(490) derived from nLw(660) could not be used to estimate NIR nLw for turbid waters with Kd(490) > 5.0 m$^{-1}$ [13]. The NASA-STD algorithm establishes the relations between Rrs and Chla indirectly, and the YS region is characterized by high concentrations of total suspended matter and CDOM. Therefore, the iterative NIR correction might not be a complete fit for the YS region. The MUMM method that assumes spatial homogeneity of the ratio of Rrs and $\rho_A$ might not always be suitable for the YS region in summer [9]. Some studies have shown that the ratio of Lw is not usually constant in the NIR bands for turbid coastal waters [65]. This ratio can vary in the range 1.0–2.0 with a change in water turbidity [67], which could be described as a function of the absorption coefficient of pure water in the NIR bands and the suspended particulate matter [16,68].

The aerosol models selected by the four algorithms are different. Aerosol concentrations and components are difficult to determine because of their high spatiotemporal variation [62]. The intermediate products for AC, i.e., aerosol products, can be substantially different with different aerosol models. The four AC algorithms use the two NIR bands to select the two most suitable models, but the aerosol products generated from SeaDASv7.5 (NASA-STD and MUMM algorithms) consider 80 aerosol models with eight relative humidity values (30%, 50%, 70%, 75%, 80%, 85%, 90%, and 95%) and 10 aerosol particle size distributions for each value of relative humidity [12]. Conversely, the KOSC-STD method uses the simplified nine aerosol models from GW94: Oceanic with relative humidity of 99% (O99), Maritime (M50, M70, M90, and M99), Coastal (C50 and C70), and Tropospheric (T50 and T90) [69]. The hypothesis used in SeaDASv7.5, which is based on using single scattering to determine

the aerosol models (weighting factor) and to apply the weighting factor directly to all bands, neglects the nonlinear relationship not only between suspended sediments and multiple-scattering aerosol reflectance but also between the interband relations of multiple scattering. To avoid the unsuitability of this condition, the spectral relationships in the aerosol multiple-scattering reflectance contributions are updated in GDPSv2.0 [70], which are improved by 1.1% in comparison with suspended sediments. Moreover, aerosol models used in SeaDASv7.5 and GPDSv2.0 are non- or weakly-absorbing aerosols that might not be completely representative of the aerosols over the YS region [observed values of AOT(870) are >0.3 with AE values of around 1.5].

Slot effects are evident features of GOCI data. An L1B GOCI image consists of 16 slot images. The slot border stray-light effect is a particular artifact of the GOCI optical system that creates clear inconsistency between adjacent slots. It primarily has an impact on the AE [29]. To minimize its influence, an image-based method called the minimum noise fraction transform has been proposed for the partial removal of the slot border [71]. In the future, this method will be applied in GDPS Software. However, NASA's Ocean Biology Processing Group has still not considered using the method to mitigate the problem. Based on our study, it is considered that slot effects have obvious influence on the images (Figure 10).

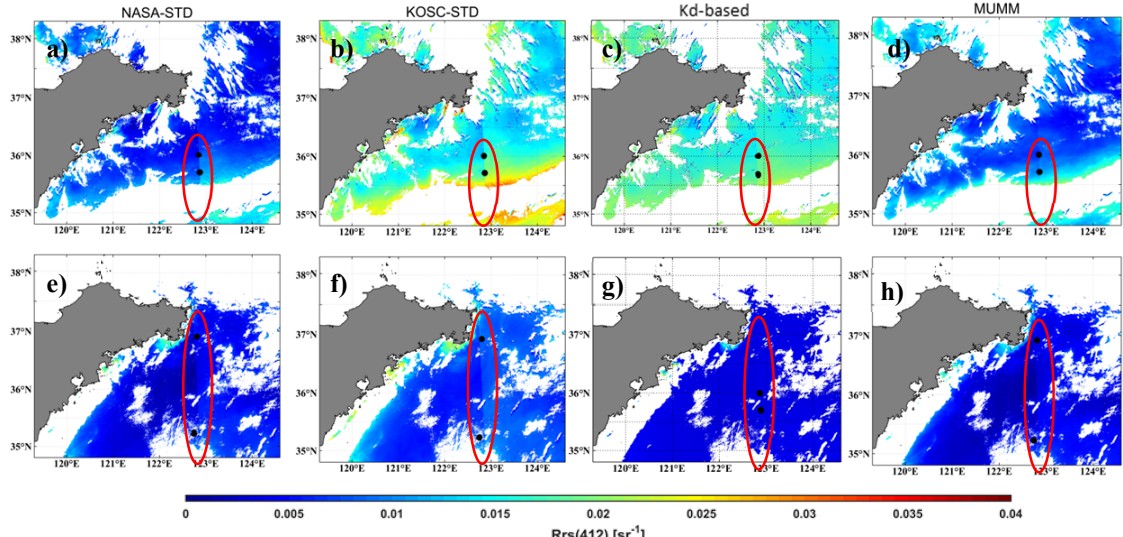

**Figure 10.** Red ellipses highlight slot border effects. The black dots denote the match-ups located in the slot border. Each column (left to right) represents Rrs(412) for NASA-STD, KOSC-STD, Kd-based, and MUMM algorithms, respectively. (**a**–**d**) and (**e**–**h**) depict results for 22 and 23 August 2017 (UTC = 03:00), respectively.

Vicarious calibration (VC) of the GOCI system is conducted to ensure that a relative radiometric calibration to GOCI minimizes uncertainties in the derived remote sensing reflectance products. The system VC gains for GOCI are slightly different between NASA and KIOST/KOSC. For NASA, it is applied as a one-time gain without any in-depth or long-term evaluation [72], whereas the default VC gains processed by SeaDASv7.5 are 0.9726, 0.9520, 0.9258, 0.8974, 0.9007, 0.8719, 0.9430, and 1.0 for each GOCI band [72]. In the present GDPSv2.0, the VC gains for 412–865 nm are 1.0118, 0.9954, 0.9715, 0.93431, 0.9596, 0.9669, 0.96125, and 1.0 [73,74]. The influence of VC on AC between KIOST/KOSC and NASA will be evaluated in future research. The MUMM and the Kd-based algorithms were not calibrated vicariously in this study (thus, it is beyond the scope of this article).

## 6. Conclusions

This study evaluated the performances of the NASA-STD, the KOSC-STD, the Kd-based, and the MUMM algorithms for GOCI over the YS at three observation times (02:16, 03:16, and 04:16 UTC)

with temporal windows of ±1 and ±3 h using in situ aerosol optical measurements (AOT and AE), Rrs, and Chla data from shipborne observations. Our quantitative results indicate that the KOSC-STD method was the algorithm most appropriate for deriving Rrs and aerosol indices in summer over the YS region with ±1 and ±3 h windows. The Kd-based NIR correction algorithm was the second most accurate algorithm, and the MUMM method was the least accurate. To interpret the temporal changes of GOCI-derived Rrs and AOT products, the results suggest that Rrs and AOT products were closer to in situ values at 04:16 UTC.

In terms of Rrs, all algorithms showed better performance at 490 and 555 nm for all stations at all GOCI observing times and low accuracies in the blue (412 nm) and the NIR bands (745–865 nm). For the band ratio of Rrs commonly used to estimate Chla, the use of Rrs(490)/Rrs(555) is recommended for the YS region rather than Rrs(443)/Rrs(555). For Chla retrievals, the Kd-based and the KOSC-STD algorithms produced Chla concentration estimations that were most accurate; the NASA-STD and the MUMM methods did not perform well in this process in our study.

For AOT, the uncertainties of satellite-retrieved values were greatest at 443 and 865 nm. AOT(555) showed reasonably satisfactory agreement with in situ AOT. However, none of the algorithms properly estimated AE, which means further work is needed to expand consideration of aerosol type and to deal with the influence of moderately and strongly absorbing aerosols.

Further improvements and optimization for AC in coastal turbid waters could be undertaken as follows. (1) The choice of NIR correction models has a crucial effect on AC accuracy because such models are often based on several hypotheses or relationships between interband Lw. The classification of water type in coastal oceans [65,75,76] will help to refine the bio-optical coefficients in NIR correction models; however, some studies have classified China's coastal waters into several types. (2) The AC process requires a considerable amount of in situ aerosol data (e.g., aerosol particle size distribution and index of refraction) [77] to validate and develop regional AC algorithms. A large network of ground stations such as AERONET [55], which relies on CE-318 automated sun photometers, could provide wide coverage; however, there are few such stations distributed in Chinese coastal waters at present. Therefore, it is urgently required that an aerosol observation network be established over China's coast. (3) The latest demands regarding sensors have been for the addition of UV bands because such bands are sensitive to aerosol absorption [78]. Consequently, GOCI-II will have UV bands [79]. (4) We also need realistic moderately and strongly absorbing aerosol models for generating realistic aerosol look-up tables. (5) One of the most important future tasks is to tackle the limited number of Rrs match-ups to assess the performance of AC over coastal waters with several different optical constituents. Considerable efforts are being made to enable the instruments to obtain continuous Rrs data near Chinese coastal waters. Additionally, the present algorithms should be validated with long-term observations covering different seasons and different coastal areas.

**Author Contributions:** Conceptualization, X.H., J.Z. and B.H.; methodology, X.H., C.J. and B.H.; software, X.H., Z.T. and Y.Z.; validation, X.H., Z.T. and T.L.; formal analysis, J.Z., C.J. and J.L.; writing—original draft preparation, X.H. and Z.T.; writing—review and editing, B.H. and C.J.; funding acquisition, J.Z. and B.H.

**Funding:** This work was funded by the National Key Research and Development Program of China (2016YFC1400906, 2016YFC1400903) and Operational Support Service System For Natural Resources Satellite Remote Sensing.

**Acknowledgments:** We thank KIOST/KOSC and OBPG/NASA for the provision of GOCI data and support with the GDPS and SeaDAS software, respectively. We are grateful to both Dr. Jae-Hyun Ahn and Dr. Myungje Choi for providing KIOST/KOSC official aerosol products of GOCI and for their advices and help. All members of the crew of the C/V *Haili* are highly acknowledged for their hard work in collecting and analyzing the in situ measurements. We also thank the reviewer for giving us constructive comments. Cédric Jamet visited the National Ocean Technology Center through funding from the Université du Littoral-Côte d'Opale. We thank James Buxton MSc from Liwen Bianji, Edanz Group China (www.liwenbianji.cn./ac), for editing the English text of this manuscript.

**Conflicts of Interest:** The authors declare no conflict of interest.

## Appendix A

Figure A1 represent the spectra of Rrs over five subregions. According to these figures, the peaks of the spectra for PSD, YT, and TZ are around 555 nm and around 490 nm for WH. For CWM, the spectra of Rrs decrease rapidly when the wavelength increases in most of the stations.

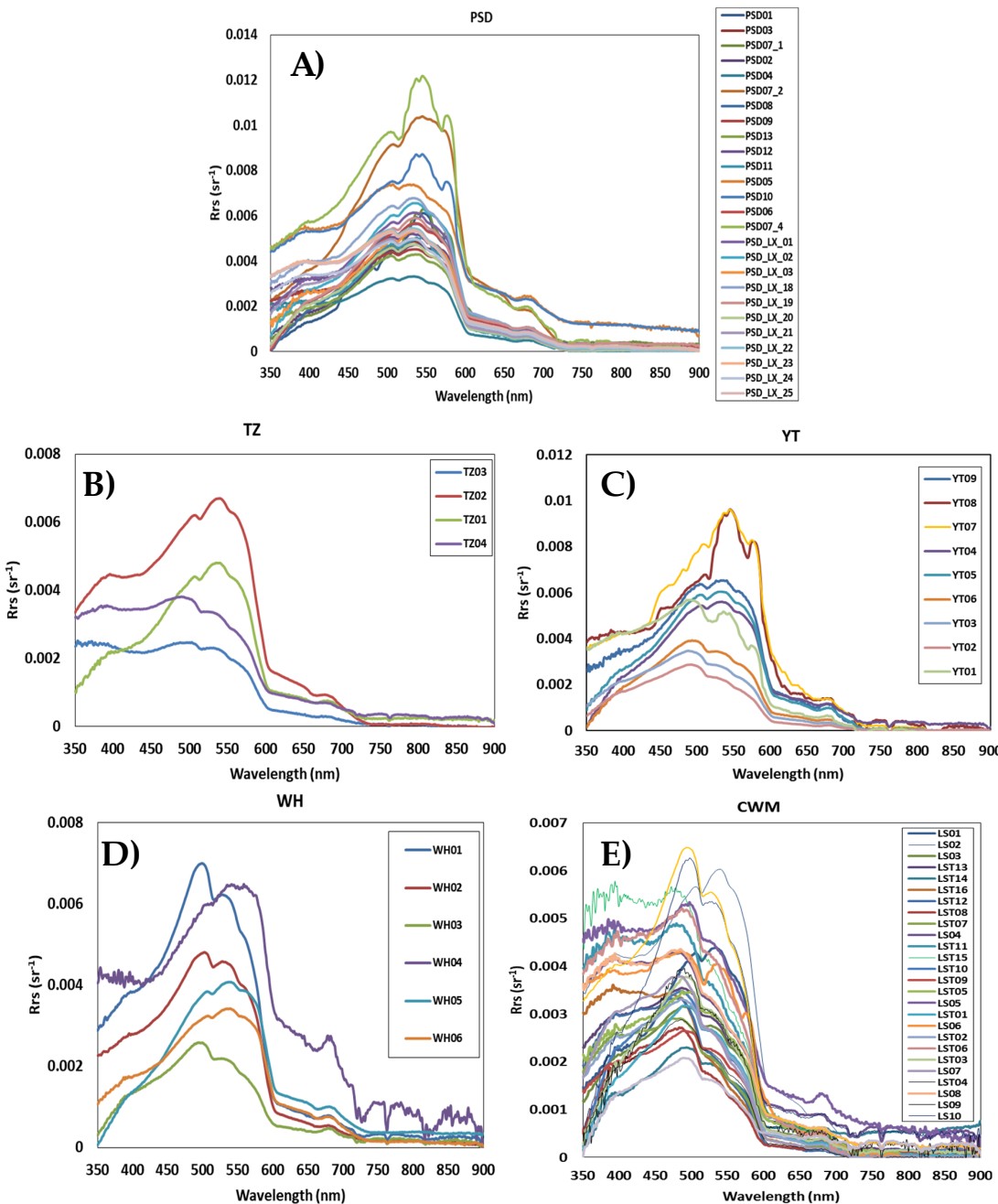

**Figure A1.** (**A**–**E**) In situ spectra of Rrs in five subregions over the YS.

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
