# Peer review of "Evaluation of Four Atmospheric Correction Algorithms for GOCI Images over the Yellow Sea"

_remotesensing, doi:10.3390/rs11141631_

Round 1

Reviewer 1 Report

Manuscript remotesensing-500935

 Evaluation of four atmospheric correction algorithms 2 for GOCI images over the Yellow Sea 

Xiaocan Huang et al.

The manuscript has been significantly improved in comparison to the first version. 

There are some problems with formatting the Tables, and I have found several typos.

For example:

Line 17 should be ‘in summer’

Line 107, please delete ‘in brief’

Line 298- formatting

Line 470- formatting

In conclusion, I think that this paper can be accepted for publication, after this corrections are made.

Author Response

Thank you so much.

We have revised the format of tables (Table 1 in the previous manuscript is divided into Table 1 to Table 4 in this new manuscript) and languages.

Reviewer 2 Report

Dear authors,

I believe the study has improved significantly. Most of my questions were answered and I deeply encourage the authors to include the figures of Rrs in each station in the appendix of the manuscript as suggested in the response.

A first minor suggestion is to improve Figure 2-C since it is very hard to see the legend and impossible to distinguish between stations. I would also suggest to change the color of the lines.

A second suggestion is to include some other methods in the introduction as a review of the literature. ACOLITE for example is in fact a software for Landsat and Sentinel but the method is described in  Vanhellemont and Ruddick (Vanhellemont, Q., Ruddick, K., 2014. Turbid wakes associated with offshore wind turbines observed with Landsat 8. Remote Sens. Environ. 145, 105–115.), and as it focus in turbid coastal waters, can be tested in future GOCI studies.

Regards 

Author Response

We have revised the legend of Figure 2C (in bold) to see clearly. And added the spectra of Rrs in the five sub regions in Appendix. 

Thank you so much!

Reviewer 3 Report

Thanks to the author of the manuscript for such a major change, this version is much better than the previous version. I recommend it for publication.

Author Response

Thank you so much! 

This manuscript is a resubmission of an earlier submission. The following is a list of the peer review reports and author responses from that submission.

Round 1

Reviewer 1 Report

Review 

Manuscript Number: 474283

Title: Evaluation of three atmospheric correction algorithms for GOCI images over the Yellow Sea

Authors: Xiaocan Huang et al.

I have read with interest this manuscript. The accuracy of satellite-derived ocean color products in coastal ocean depends on atmospheric correction algorithms. These algorithms aim to remove from satellite-recorded radiances the contribution of the atmosphere and surface reflection, in order to quantify water-leaving radiances (Lw). Water-leaving radiances are then used to derive ocean color products such as the chlorophyll concentration. The problem is that atmosphere may contribute about 90% or more to the TOA radiance measured by a satellite sensor and that the Lw needs to be derived with errors less than 5 %. It is particularly difficult to perform precise atmospheric corrections in coastal regions, where both the water and the aerosols are optically more complex than in open ocean regions. Over the recent years different approaches were proposed to overcome this problem, but this is still an active area of research. The paper is therefore of interest and the problem addressed is timely.

After reading the manuscript I have several concerns:  

1) The language needs to be corrected by a professional proofreader because it is sometimes hard to follow the text.  I have marked in the attached pdf copy of the manuscript many places that require corrections, but I am sure that more corrections are needed.

2) The authors should pay attention that all the abbreviations are explained when they appear in the text for the first time. The same goes for all the symbols in the mathematical expressions. Also, please pay attention that you use scientific language, for example see comment for line 139.

3) Please correct Figures, in particular Figures 2  and 8 (see comment in the pdf). Also-add a) b) c) d) and use this to improve Figure Captions. Perhaps you could find a way to distinguish 1-hour and 3-hour time intervals in the scatterplots? (for example filled and not filled markers of the same color?).

4) Regarding the matchup procedures- I am afraid that 3 hours is too long time interval, considering that this region is highly variable. It is OK to show both 1 hour and 3 hours comparisons, but you need to point this out in the discussion, that 3 hours is significantly affected by the in situ variability (this conclusion is supported by your results).

Unfortunately, there are not that many matchups. Could you increase your data base, by adding some older data or data from SeaBass (NASA)? Could you check if you can apply similar data processing for MODIS?

5) Could you show what is the impact on the errors in AC on derived Chl concentrations?

6) Tables with statistics would be better if you mark in bold the best results. Always indicate in the Tables if they are based on 3 hour or 1 hour time interval. Table 3 should indicate how many negative retrieval were obtained.

7) If you want to compare quantitatively the statistical results - you should show the confidence intervals. This way we would know if the differences between the results from different algorithms are statistically significant.

8) I would like to see all the results based on matchups with 1-hour time interval.

All these corrections should be made before the English is corrected.

Author Response

As for your questions, the answers are following:

1)       The language needs to be corrected by a professional proofreader because it is sometimes hard to follow the text.  I have marked in the attached pdf copy of the manuscript many places that require corrections, but I am sure that more corrections are needed.

We have revised the language according to your pdf. The manuscript has also been revised by a professional proofreader for the language.

2)    The authors should pay attention that all the abbreviations are explained when they appear in the text for the first time. The same goes for all the symbols in the mathematical expressions. Also, please pay attention that you use scientific language, for example see comment for line 139.

We have read carefully our manuscript and now all symbols and abbreviations designed a specific quantity.

3)    Please correct Figures, in particular Figures 2  and 8 (see comment in the pdf). Also-add a) b) c) d) and use this to improve Figure Captions. Perhaps you could find a way to distinguish 1-hour and 3-hour time intervals in the scatterplots? (for example filled and not filled markers of the same color?).

We have added a) b) c) d) into each sub-figure.

In order to distinguish 1-hour and 3-hour time-windows, we have added the statistical results for Rrs and AOT, respectively (We have considered using scatterplots to distinguish them, but there are four algorithms for 2 time intervals. It's difficult to properly observe the differences when plotting all algorithms altogether. Therefore we decided to put results as a table.: Table 1 to Table 4. In each table, the italic of each indice represents 3-hour, while the results in the brackets correspond to 1-hour.

4)    Regarding the matchup procedures- I am afraid that 3 hours is too long time interval, considering that this region is highly variable. It is OK to show both 1 hour and 3 hours comparisons, but you need to point this out in the discussion, that 3 hours is significantly affected by the in situ variability (this conclusion is supported by your results).

Table 1,2,3,4 showed the results of 1 hour and 3 hours. And we added the discussion about 1-hour and 3-hour in the part of results and discussion. Within a ±3h match-up window, of the 78 available in situ Rrs measurements, 30, 29, 31, and 32 match-ups for Rrs(555) are available for the NASA-STD, KOSC-STD, Kd-based, and MUMM algorithms, respectively. The number of match-ups for each algorithm is greater with a time- window of ±3 h than for a time- window of ±1 h: 10, 10, 10, and 11 match-ups. The number of match-ups for AOT with ±3 h and ±1 h windows is similar to Rrs(555). Soin this manuscript, we take ±3 h into account.

Unfortunately, there are not that many matchups. Could you increase your data base, by adding some older data or data from SeaBass (NASA)? Could you check if you can apply similar data processing for MODIS?

There are no stations near Yellow Sea coastal regions from SeaBass. Our team had done the cruise experiment in 2003 over Yellow Sea and East China Sea, but GOCI was not launched yet. So, we just used the field data acquired in 2017 for our study.

Numerous articles have been already published on validation of atmospheric correction or bio-optical algorithms with less match-ups for the validation of several parameters (e.g. Oo et al, 2008; Cui et al., 2010; Guanter et al., 2010; Beltran-Abaunza et al., 2014; Cui et al., 2014; Zhao et al., 2014; Delgado et al., 2015; Jaelani et al., 2015; De Keukelaere et al., 2018). It is very difficult to get a high number of match-ups. We consider, as a rule, that there is a ratio of 10 for 1 between the number of in-situ measurements and the number of match-ups. So having more match-ups either means to go more often in the sea over a shorter period of time or waiting longer for publishing. We believe our dataset, even small, represents a wide variety of water types in the Yellow Sea.

5)    Could you show what is the impact on the errors in AC on derived Chl concentrations?

We added information on sub-section 2.2, 4.3 and discussion.

6)    Tables with statistics would be better if you mark in bold the best results. Always indicate in the Tables if they are based on 3 hour or 1 hour time interval. Table 3 should indicate how many negative retrieval were obtained.

We have marked in bold the best results. And in each table, we exhibit 1 hour and 3 hour time-windows.

We exchange the introduction sequence of Rrs and the aerosol optical properties (in the revised manuscript, we introduce Rrs first, and then AOT). So, Table 1 about Rrs has indicated the negative numbers.

7)    If you want to compare quantitatively the statistical results - you should show the confidence intervals. This way we would know if the differences between the results from different algorithms are statistically significant.

We analyse the confidence intervals for Chla in Table 4. It is showed that the KOSC and Kd-based algorithm are the best algorithms to retrieve Chla.

8)    I would like to see all the results based on matchups with 1-hour time interval.

Table 1 to Table 4

All these corrections should be made before the English is corrected.

The corrections have been made before submitting the manuscript to a professional proorfreader.

Reviewer 2 Report

The paper” Evaluation of three atmospheric correction algorithms for GOCI images over the Yellow Sea” presents an important evaluation on Atmospheric correction (AC) algorithms for GOCI images, particularly for scientists which investigate the yellow sea region. Although the topic is important I believe that the manuscript needs further improvement before publication.

The main aspect, which I believe, needs to be addressed is the field dataset. Both, Rrs and AOD data should be presented separately so the reader can understand the variability of those parameters within the study area. Particularly, for Rrs data, the factor used by Mobley (1999) (which is not actually the fresnel reflectance) to correct for gilnt effects can deeply impact the values, particularly in the infrared and blue regions. The table provided by Mobley 2015 with new factor accounting for polarization should be used to correct each Rrs field spectra and guarantee the quality of the data. The factor can be drastically different from the 0.028 depending on wind speed. Since all comparison has been done assuming no errors in the field data it is difficult to believe that the all errors are due to the AC algorithms.

Also, I could see that field data was collected in 6 different regions of the yellow sea. What are the Rrs differences among those regions ? Are there optical differences based on the variability of optical components ? I believe that  both, field data and the results for AC algorithms should be presented for the different regions and therefore the reader can understand if the performance of all 3 tested AC algorithms is suitable for each region.

Regarding the AC algorithms, I would expect more discussion, maybe in the Introduction on the comparison to other algorithms like the one used in ACOLITE or in sen2cor which are vastly discussed in the literature for Landsat and Sentinel 2/3 remote sensing data. Are those algorithms similar to the ones tested in this study ? Can they also be applied for GOCI ?

At last, I would encourage the authors not only to compare the band ratio to but to use the Chla algorithms in the evaluation of AC algorithms. As an example, how the values of Rrs would change if and three band Chla algorithm is used ? Or an “OC5 like” applied to GOCI data. Therefore the impact of atmospheric correction could be directly related to users interested in Chla studies.

Author Response

Point 1: The paper” Evaluation of three atmospheric correction algorithms for GOCI images over the Yellow Sea” presents an important evaluation on Atmospheric correction (AC) algorithms for GOCI images, particularly for scientists which investigate the yellow sea region. Although the topic is important I believe that the manuscript needs further improvement before publication.

The main aspect, which I believe, needs to be addressed is the field dataset. Both, Rrs and AOD data should be presented separately so the reader can understand the variability of those parameters within the study area. Particularly, for Rrs data, the factor used by Mobley (1999) (which is not actually the fresnel reflectance) to correct for gilnt effects can deeply impact the values, particularly in the infrared and blue regions. The table provided by Mobley 2015 with new factor accounting for polarization should be used to correct each Rrs field spectra and guarantee the quality of the data. The factor can be drastically different from the 0.028 depending on wind speed. Since all comparison has been done assuming no errors in the field data it is difficult to believe that the all errors are due to the AC algorithms.

Response 1: We agree with the reviewer that the in-situ measurements can also have errors. For the estimation of Rrs, ρs(λ) was retrieved by means of the non-linear spectral optimization method and a bio-optical model for each station as done in Cui et al., 2013 and Lee et al.,2010, instead of the constant 0.028. We have confirmed this value for each in situ station with the data processor in our team.

Point 2: Also, I could see that field data was collected in 6 different regions of the yellow sea. What are the Rrs differences among those regions ? Are there optical differences based on the variability of optical components ? I believe that  both, field data and the results for AC algorithms should be presented for the different regions and therefore the reader can understand if the performance of all 3 tested AC algorithms is suitable for each region.

Response 2: We marked these 6 different regions of the Yellow Sea just to distinguish the different field data ,the number of match-up spectra for each sub-region is quite low so we couldn't perform any significative statistical analyses. Therefore, in this study, we do not evaluate the AC for these 6 different regions (The spectrum of Rrs over each sub-region are in the appendix).

In the figures, the peak of spectrum for PSD, YT, TZ are around at 555nm, for WH are around at 490nm. For Cold Water Mass, most of stations belong to the Case I water spectrum, the Rrs decreases with the bands increase.

Point 3: Regarding the AC algorithms, I would expect more discussion, maybe in the Introduction on the comparison to other algorithms like the one used in ACOLITE or in sen2cor which are vastly discussed in the literature for Landsat and Sentinel 2/3 remote sensing data. Are those algorithms similar to the ones tested in this study ? Can they also be applied for GOCI ?

Response 3: The software ACOLITE or sen2cor is just for Landsat and Sentinel-2, there are Integration software with fixed code. GOCI has its band response function, so we cannot use these software directly.

Point 4: At last, I would encourage the authors not only to compare the band ratio to but to use the Chla algorithms in the evaluation of AC algorithms. As an example, how the values of Rrs would change if and three band Chla algorithm is used ? Or an “OC5 like” applied to GOCI data. Therefore the impact of atmospheric correction could be directly related to users interested in Chla studies.

Response 4: We used OC3G algorithms to derive Chla in sub-section 2.2.

Reviewer 3 Report

This paper is well organized and written. However, the results are not impressive. The three algorithms evaluated in this paper were developed many years ago and have already been well assessed, especially for MUMM and the NASA standard. The NASA standard algorithm is inapplicable to coastal turbid waters, of which optical properties are dominated by TSM instead of phytoplankton. The two assumptions used in MUMM can result in large deviations when applied to coastal waters. Actually, this algorithm has been improved for turbid coastal waters by some researchers. The algorithm implemented in GDPS selects aerosol model based on the nonlinear empirical relationship of spectral water reflectance between red (660 nm) and two NIR bands (745 and 865 nm). Therefore, it is not surprising that GDPS performs best in the study area. I think it makes little sense to compare these three algorithms. It would have been interesting to evaluate other new or improved atmospheric correction algorithms proposed in recent years. For example, the algorithm based on the Kd proposed by Wang et al. (2013).

Author Response

Point: This paper is well organized and written. However, the results are not impressive. The three algorithms evaluated in this paper were developed many years ago and have already been well assessed, especially for MUMM and the NASA standard. The NASA standard algorithm is inapplicable to coastal turbid waters, of which optical properties are dominated by TSM instead of phytoplankton. The two assumptions used in MUMM can result in large deviations when applied to coastal waters. Actually, this algorithm has been improved for turbid coastal waters by some researchers. The algorithm implemented in GDPS selects aerosol model based on the nonlinear empirical relationship of spectral water reflectance between red (660 nm) and two NIR bands (745 and 865 nm). Therefore, it is not surprising that GDPS performs best in the study area. I think it makes little sense to compare these three algorithms. It would have been interesting to evaluate other new or improved atmospheric correction algorithms proposed in recent years. For example, the algorithm based on the Kd proposed by Wang et al. (2013).

Response:  We don't agree with the reviewers about the NASA and MUMM algorithms. NASA standard algorithm include turbid waters for the past ten years (Stumpf et al., 2003; Bailey et al., 2010). Some publications showed the relevance of using this algorithm for studying coastal waters. It's the same for the MUMM algorithm even if it has been known that it has limitations for very turbid waters (Goyens et al., 2013).

The algorithm of Wang et al. (2013) based Kd(490) has been added now in our manuscript.